# Coordination between nucleotide excision repair and specialized polymerase DnaE2 action enables DNA damage survival in non-replicating bacteria

Asha Mary Joseph[1], Saheli Daw[1], Ismath Sadhir[1,2], Anjana Badrinarayanan[1]*

[1]National Centre for Biological Sciences - Tata Institute of Fundamental Research, Bangalore, India; [2]Max Planck Institute for Terrestrial Microbiology, LOEWE Centre for Synthetic Microbiology (SYNMIKRO), Marburg, Germany

**Abstract** Translesion synthesis (TLS) is a highly conserved mutagenic DNA lesion tolerance pathway, which employs specialized, low-fidelity DNA polymerases to synthesize across lesions. Current models suggest that activity of these polymerases is predominantly associated with ongoing replication, functioning either at or behind the replication fork. Here we provide evidence for DNA damage-dependent function of a specialized polymerase, DnaE2, in replication-independent conditions. We develop an assay to follow lesion repair in non-replicating *Caulobacter* and observe that components of the replication machinery localize on DNA in response to damage. These localizations persist in the absence of DnaE2 or if catalytic activity of this polymerase is mutated. Single-stranded DNA gaps for SSB binding and low-fidelity polymerase-mediated synthesis are generated by nucleotide excision repair (NER), as replisome components fail to localize in the absence of NER. This mechanism of gap-filling facilitates cell cycle restoration when cells are released into replication-permissive conditions. Thus, such cross-talk (between activity of NER and specialized polymerases in subsequent gap-filling) helps preserve genome integrity and enhances survival in a replication-independent manner.

*For correspondence:
anjana@ncbs.res.in

Competing interests: The authors declare that no competing interests exist.

## Introduction

DNA damage is a threat to genome integrity and can lead to perturbations to processes of replication and transcription. In all domains of life, bulky lesions such as those caused by UV light (cyclobutane pyrimidine dimers, CPD and to a lesser extent 6,4 photoproducts, 6-4PP) are predominantly repaired by nucleotide excision repair (NER) (*Boyce and Howard-Flanders, 1964*; *Chatterjee and Walker, 2017*; *Kisker et al., 2013*). This pathway can function in global genomic repair (GGR) via surveilling the DNA double-helix for distortions or more specifically via transcription-coupled repair (TCR) (*Kisker et al., 2013*). The main steps of NER involve lesion detection followed by incision of few bases upstream and downstream of the lesion, resulting in removal of a short stretch of single-stranded DNA (ssDNA). In some cases, NER can also result in generation of longer patches of ssDNA (*Cooper, 1982*). ssDNA gaps are then filled by synthesis from a DNA polymerase (*Kisker et al., 2013*; *Sancar and Rupp, 1983*). While the NER-mediated damage removal pathway is largely error-free, lesions encountered by the replication machinery (e.g., CPDs, 6-4PPs, and cross-links such as those generated by antibiotics including mitomycin C [MMC]) can also be dealt with via error-prone translesion synthesis (TLS) (*Chatterjee and Walker, 2017*; *Fuchs and Fujii, 2013*; *Fujii and Fuchs, 2004*).

TLS employs low-fidelity polymerases to synthesize across DNA lesions, with increased likelihood of mutagenesis during this process (*Fuchs and Fujii, 2013*; *Galhardo et al., 2005*; *Kato and*

*Shinoura, 1977*; *Nohmi et al., 1988*; *Warner et al., 2010*). In most bacteria, expression of these polymerases is regulated by the SOS response, which is activated by the RecA-nucleoprotein filament under DNA damage (*Baharoglu and Mazel, 2014*). Currently, most of our understanding about TLS comes from studies on specialized Y-family polymerases of *E. coli*, DinB (PolIV) and UmuDC (PolV), both of which function in DNA lesion tolerance and contribute to mutagenesis in several bacterial systems (*Kato and Shinoura, 1977*; *Nohmi et al., 1988*; *Steinborn, 1978*; *Sung et al., 2003*; *Wagner et al., 1999*). In addition, PolV has also been implicated in RecA-dependent post-replicative gap-filling activity (*Isogawa et al., 2018*). In contrast to *E. coli*, *Caulobacter crescentus* as well as other bacteria including *Mycobacterium sp.* and *Pseudomonas sp.* encode an alternate, SOS-inducible error-prone polymerase, DnaE2 (*Boshoff et al., 2003*; *Galhardo et al., 2005*; *Jatsenko et al., 2017*; *Warner et al., 2010*). DnaE2 is highly conserved and is mutually exclusive with PolV in occurrence. In the limited organisms where DnaE2 has been studied so far, it is the primary TLS polymerase and the only contributor to damage-induced mutagenesis (*Alves et al., 2017*; *Galhardo et al., 2005*; *Warner et al., 2010*). In contrast to PolV, DnaE2 is thought to preferentially act on MMC-induced damage, where it contributes to all induced mutagenesis observed (*Galhardo et al., 2005*). In case of UV, there are still uncharacterized mechanisms that can contribute to damage tolerance and mutagenesis that are independent of DnaE2 (*Galhardo et al., 2005*). DnaE2 co-occurs with ImuB, a protein that carries a β-clamp binding motif, and is thought to act as a bridge between DnaE2 and the replisome (*Warner et al., 2010*). Unlike *E. coli*, where activities of PolIV and PolV are well-studied, in vivo investigations of DnaE2 function in damage tolerance and cellular survival are limited. This becomes particularly important, given the emerging evidences across domains of life ascribing diverse functions to these low-fidelity polymerases beyond their canonical function of replication-associated lesion bypass (*Joseph and Badrinarayanan, 2020*). Indeed, such polymerases are also referred to as 'specialized polymerases' (*Fujii and Fuchs, 2020*) so as to consider these broader functions.

Since error-prone polymerases can synthesize DNA and their activity is mediated by interaction with the β-clamp of the replisome (*Bunting et al., 2003*; *Chang et al., 2019*; *Fujii and Fuchs, 2004*; *Thrall et al., 2017*; *Wagner et al., 2009*; *Warner et al., 2010*), action of these polymerases has mostly been studied in the context of replicating cells, as a mechanism that facilitates continued DNA synthesis by acting at or behind the replication fork (*Chang et al., 2019*; *Chang et al., 2020*; *Indiani et al., 2005*; *Jeiranian et al., 2013*; *Marians, 2018*). In addition to replication-associated lesion tolerance, some studies have proposed the possibility of error-prone synthesis in a manner that is replication-independent (*Janel-Bintz et al., 2017*; *Kozmin and Jinks-Robertson, 2013*). This is supported by observations that cells can undergo stationary phase mutagenesis that is dependent on action of error-prone polymerases (*Bull et al., 2001*; *Corzett et al., 2013*; *Janel-Bintz et al., 2017*; *Sung et al., 2003*; *Yeiser et al., 2002*). Microscopy-based approaches have also provided evidence in line with the idea that tolerance or gap-filling could occur outside the context of the replication fork in *E. coli*, as replisome components, such as the β-clamp, as well as specialized polymerases (PolIV and PolV) were found to localize away from the fork in response to DNA damage (*Henrikus et al., 2018*; *Robinson et al., 2015*; *Soubry et al., 2019*; *Thrall et al., 2017*). Furthermore, while originally considered as distinct mechanisms of repair (damage tolerance vs. damage removal), recent studies also suggest cross-talk between specialized polymerases and NER in *E. coli*, yeast, and human cells (*Giannattasio et al., 2010*; *Janel-Bintz et al., 2017*; *Kozmin and Jinks-Robertson, 2013*; *Sertic et al., 2018*). Indeed, long-standing observations suggest that NER can be mutagenic under certain conditions in *E. coli*, in a manner that is dependent on RecA (*Bridges and Mottershead, 1971*; *Cohen-Fix and Livneh, 1994*; *Nishioka and Doudney, 1969*). However, the mechanistic basis of this process in replication-independent conditions and conservation of the same across bacteria that encode diverse specialized polymerases remains to be elucidated. For example, unlike *E. coli*, several bacterial systems undergo nonoverlapping cycles of DNA replication and have distinct cell cycle phases with no ongoing DNA synthesis. The relevance of lesion correction or gap-filling for genome integrity maintenance in the absence of an active replication fork (such as in non-replicating swarming cells) is relatively less explored, especially in bacterial contexts.

To probe the in vivo mechanism and understand the impact of error-prone polymerase function in non-replicating bacteria, we investigated lesion repair in *Caulobacter crescentus* swarmer cells. *Caulobacter* is well-suited to study activity of these specialized polymerases due to its distinct cell cycle. Every cell division gives rise to two different cell types: a stalked and a swarmer cell. While the

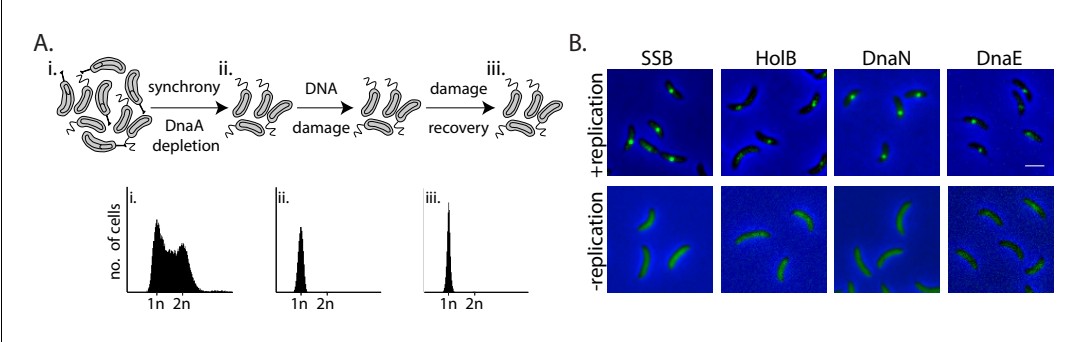

**Figure 1.** Monitoring mechanisms of DNA lesion repair in non-replicating bacteria. (**A**) Above: Schematic of experimental setup used to isolate non-replicating *Caulobacter* swarmer cells to monitor DNA lesion repair and tolerance independent of ongoing replication. Cells were treated with DNA damage (30 min mitomycin C [MMC] or UV at specified doses), after which damage was removed and cells were allowed to grow in fresh media (damage recovery), without ongoing replication. Below: Flow cytometry profiles show DNA content in an asynchronous population (**i**), synchronized non-replicating swarmer cells before (**ii**) and after DNA damage recovery (**iii**). (**B**) Representative images of *Caulobacter* cells with fluorescently-tagged replisome components (SSB-YFP, HolB-YFP, DnaN-YFP, or DnaE-mNG) in replicating or non-replicating conditions, without DNA damage (scale bar is 2 μm here and in all other images).

The online version of this article includes the following source data and figure supplement(s) for figure 1:

**Figure supplement 1.** Characterization of strains carrying fluorescently-tagged replisome components.

**Figure supplement 1—source data 1.** Source data related to panels in *Figure 1—figure supplement 1*.

stalked cell initiates replication soon after division, a swarmer cell must differentiate into a stalked cell before replication reinitiation (*Schrader and Shapiro, 2015*) and hence swarmers can represent a pool of naturally occurring non-replicating cells in the environment. Under laboratory conditions, these swarmer cells can be isolated via density-gradient centrifugation and replication initiation can be inhibited, resulting in a population of non-replicating cells with a single chromosome (*Badrinarayanan et al., 2015*; *Schrader and Shapiro, 2015*). Using this non-replicating system, we followed DNA damage repair with lesion-inducing agents via live-cell fluorescence microscopy. We show that low-fidelity polymerase DnaE2 functions in gap-filling damaged DNA in non-replicating cells. This is facilitated by de novo loading of replisome components (SSB, HolB [part of the clamp loader complex], β-clamp, and replicative polymerase) at long ssDNA gaps likely generated by a subset of NER events. We find that this form of gap-filling in non-replicating cells promotes cell cycle restoration and cell division, upon release into replication-permissive conditions. Our study provides in vivo evidence for a novel function of DnaE2 that is spatially and temporally separated from the active replication fork. Given that DNA damage can occur in any cell type whether actively replicating or not, coordinated activity of NER and low-fidelity polymerases can serve as a potential mechanism through which non-replicating cells such as bacteria in stationary phase or cells in other differentiated phases increase their chances of survival under damage.

## Results

### Monitoring mechanisms of DNA lesion repair in non-replicating bacteria

To test whether non-replicating cells can indeed engage in lesion repair, and understand the in vivo mechanism of such activity, we used *Caulobacter crescentus* swarmer cells as our model system. We regulated the state of replication so as to ensure that swarmer cells, with a single chromosome, do not initiate replication (and hence prevent possibility of recombination-based repair) by utilizing a previously described system to control the expression of the replication initiation regulator, *dnaA*, from an isopropyl β-D-1-thiogalactopyranoside (IPTG) inducible promoter (*Badrinarayanan et al., 2015*). In our experimental setup, we first depleted cells of DnaA for one generation of growth, followed by synchronization to isolate non-replicating swarmer cells (*Figure 1A*, top panel). Flow

cytometry profiles of cells confirmed the presence of a single chromosome during the course of the entire experiment (*Figure 1A*, bottom panel).

Given the requirement of the β-clamp for activity of specialized polymerases and evidence for damage-dependent changes in localization of replisome components such as SSB in actively replicating *E. coli* (*Chang et al., 2019*; *Henrikus et al., 2018*; *Soubry et al., 2019*; *Thrall et al., 2017*), we generated fluorescent fusions to the *Caulobacter* β-clamp (DnaN), component of the clamp-loader complex (HolB), the replicative polymerase PolIII (DnaE), and single-strand DNA binding protein (SSB) (using previously described approaches in *Caulobacter* [*Aakre et al., 2013*; *Collier and Shapiro, 2009*]; and 'Materials and methods') in order to visualize them in non-replicating swarmers. These fusions did not perturb the function of the proteins as cells displayed wild type growth dynamics in steady-state conditions (*Figure 1—figure supplement 1A and B* 'control'). They also did not show increased sensitivity to DNA damage treatment via MMC or UV (*Figure 1—figure supplement 1A and B*). The fusion proteins localized on DNA in actively replicating cells (*Figure 1B*, +replication), and as anticipated, their localizations gradually shifted from one pole to the other within one cycle of DNA replication (*Figure 1—figure supplement 1C*). These observations are in line with previous reports of replisome dynamics in several bacterial systems including *Caulobacter crescentus, Bacillus subtilis,* and *E. coli* (*Aakre et al., 2013*; *Collier and Shapiro, 2009*; *Jensen et al., 2001*; *Lemon and Grossman, 1998*; *Mangiameli et al., 2017*; *Reyes-Lamothe et al., 2008*). In contrast to actively replicating cells, replication-inhibited swarmer cells were devoid of replisome foci (*Figure 1B*), consistent with the idea that the localization of replisome components is indicative of active DNA replication.

## Replisome components are recruited to damaged DNA in non-replicating *Caulobacter* swarmer cells

Using the above described system, we treated non-replicating *Caulobacter* swarmer cells with mitomycin C (MMC) to induce DNA lesions and followed DNA damage recovery via live-cell imaging to track dynamics of the β-clamp and other replisome components (*Figure 1A*). MMC is a naturally produced antibiotic that acts predominantly on the guanine residue of DNA, making three major forms of damage: mono-adducts, intra-strand cross-links, and inter-strand cross-links (*Bargonetti et al., 2010*). In case of *Caulobacter,* it is thought that DnaE2 preferentially acts on MMC-induced damage as all mutagenesis associated with MMC treatment is mediated via action of this specialized polymerase; in absence of the polymerase, cells show high sensitivity to MMC treatment (*Galhardo et al., 2005*). To determine the range of MMC concentrations for this study, we first assessed the viable cell count for a steady-state population of wild type and Δ*dnaE2* cells across increasing concentrations of MMC treatment (0.125–2 µg/ml). We focused on a treatment range where DnaE2 essentiality was observed (*Figure 2—figure supplement 1A*) and TLS-dependent mutagenesis has been previously reported (*Galhardo et al., 2005*).

We then treated non-replicating swarmer cells with the specified doses of MMC. We found that DNA damage treatment resulted in the formation of β-clamp foci in non-replicating cells (*Figure 2A–B*). This was found to be the case for other replisome components as well (*Figure 2A–B*). The percentage of cells with damage-induced β-clamp foci increased with increasing doses of MMC. At 0.125 µg/ml MMC treatment, 9% cells had β-clamp foci, while at higher doses of 0.75 µg/ml MMC, foci were observed in 59% cells (*Figure 4—figure supplement 1C*). To further characterize the dynamics of these localizations during the course of damage recovery, we released MMC-treated non-replicating swarmers into fresh media without damage and followed the localization of replisome components over time. We maintained the block on replication initiation, thus ensuring that cells carried only a single non-replicating chromosome during the course of the experiment (*Figure 1A*). Consistent with the possibility of dissociation during recovery, we found that percentage of cells with DnaN localizations gradually decreased with time (*Figure 2C*) and across all doses of damage tested (*Figure 4—figure supplement 1C*). For example, after 30 min of 0.5 µg/ml MMC treatment, 52% cells on average had DnaN localization and at 90 min after damage removal, the number reduced to 30%. This pattern of localization after damage treatment, followed by reduction in percentage of cells with foci during recovery was also observed in the case of SSB, HolB, and DnaE (*Figure 2D*). Interestingly, we noticed that cells had more SSB localizations on average than DnaN. 14% cells had ≥2 DnaN foci after MMC treatment, while 37% cells harbored ≥2 SSB localizations. These numbers reduced with increasing time of recovery (*Figure 2D*). Assessment of the

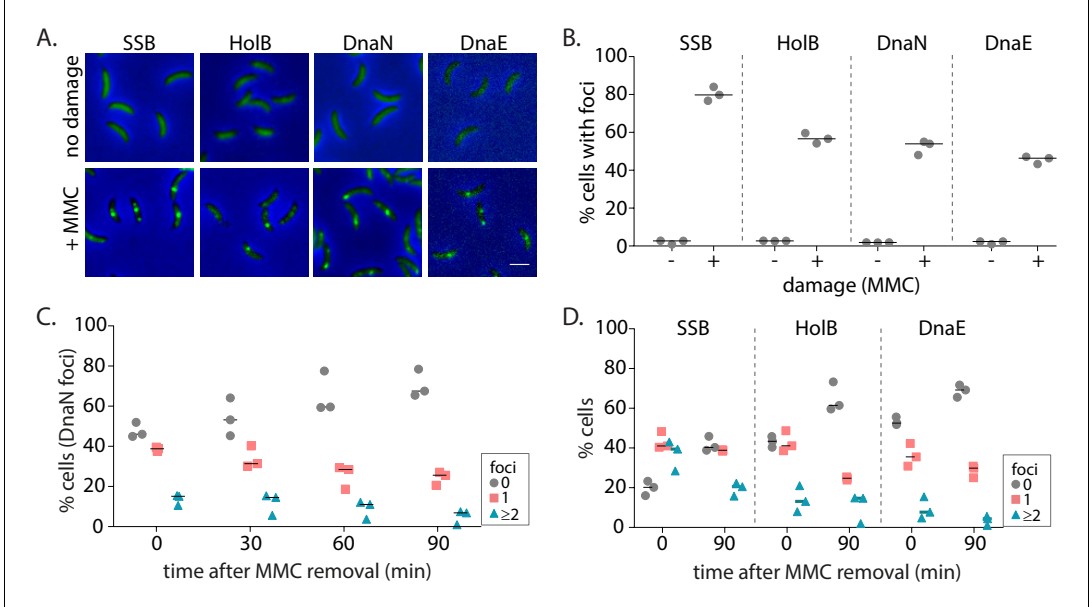

**Figure 2.** Replisome components are recruited to damaged DNA in non-replicating *Caulobacter* swarmer cells. (**A**) Representative images of non-replicating swarmer cells with fluorescently tagged replisome components (SSB-YFP, HolB-YFP, DnaN-YFP, or DnaE-mNG) with (+MMC) or without (no damage) 30 min of treatment with MMC. (**B**) Percentage cells with SSB, HolB, DnaN, or DnaE localization (foci) in non-replicating swarmers with (+) or without (-) MMC treatment (n ≥ 324 cells, three independent repeats). Dashed line represents median here and in all other graphs. (**C**) Percentage swarmer cells with 0, 1, or ≥2 DnaN foci at 0, 30, 60, and 90 min after damage removal (recovery) (n ≥ 476 cells, three independent repeats). (**D**) Percentage swarmer cells with 0, 1, or ≥2 foci of SSB, HolB, or DnaE at 0 and 90 min after damage removal (recovery) (n ≥ 324 cells, three independent repeats).

The online version of this article includes the following source data and figure supplement(s) for figure 2:

**Source data 1.** Source data related to panels in *Figure 2*.

**Figure supplement 1.** Replisome components are recruited to damaged DNA in non-replicating *Caulobacter* swarmer cells.

**Figure supplement 1—source data 1.** Source data related to panels in *Figure 2—figure supplement 1*.

extent of colocalization between DnaN and SSB further showed that 90% of DnaN foci colocalized with SSB (with distance of a DnaN focus from the nearest SSB localization being within 300 nm), while only 51% of SSB foci colocalized with DnaN (*Figure 2—figure supplement 1B and C*), suggesting that not all SSB may be associated with the β-clamp or that SSB could precede β-clamp localization.

We asked whether similar dynamics of replication machinery components were observed in the presence of a different lesion-inducing agent as well. For this, we treated cells with sub-inhibitory doses of UV radiation (*Galhardo et al., 2005* and *Figure 2—figure supplement 1D*). Exposure of cells to two doses of UV damage (75 and 150 J/m$^2$) also resulted localization and subsequent reduction in percentage of cells with replisome foci during recovery (*Figure 2—figure supplement 1E,F, G*). Taken together, these data support the idea that SSB, along with components of the PolIIIHE, including the clamp-loader, β-clamp, and the replicative polymerase, associates with DNA during damage even in the absence of ongoing replication. Decrease in localizations over time could be indicative of potential repair in non-replicating cells.

## Nucleotide excision repair (NER) generates long ssDNA gaps for localization of replisome components in non-replicating cells

How do replisome components localize in non-replicating cells? SSB foci under these conditions indicate the presence of ssDNA stretches long enough to accommodate SSB tetramers (30 nt or more) (*Bell et al., 2015*; *Lohman and Ferrari, 1994*). In replicating cells, ssDNA tracts are thought to be generated as a result of helicase activity that continues to unwind double-stranded DNA ahead of

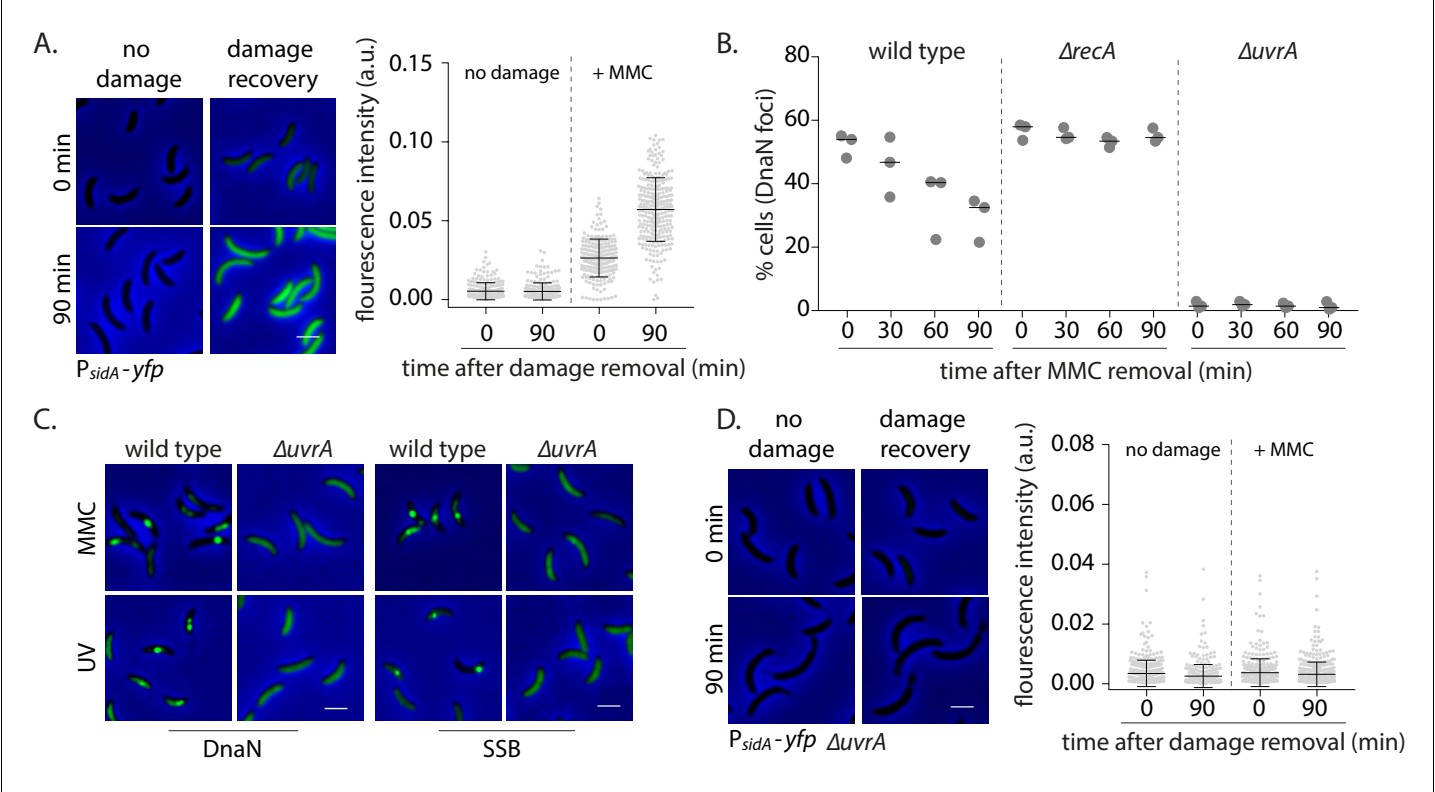

**Figure 3.** Nucleotide excision repair (NER) generates long ssDNA gaps for localization of replisome components in non-replicating cells. (**A**) SOS induction was measured by assessing the expression of *yfp* from an SOS-inducible promoter (P*sidA*-*yfp*). On the left are representative images of cells expressing the reporter at 0 or 90 min after MMC removal and control cells (no damage). On the right, total fluorescence intensity normalized to cell area is plotted for both time points for cells with or without damage treatment. Each dot represents a single cell. Mean and SD are shown in black (n ≥ 219). (**B**) Percentage wild type, Δ*recA*, or Δ*uvrA* swarmer cells with DnaN foci at 0, 30, 60, and 90 min after DNA damage recovery (n ≥ 308 cells, three independent repeats). (**C**) Representative images of wild type or Δ*uvrA* swarmer cells with SSB-YFP or DnaN-YFP, treated with MMC or UV. (**D**) As (**A**) for cells lacking *uvrA* (n ≥ 325).

The online version of this article includes the following source data and figure supplement(s) for figure 3:

**Source data 1.** Source data related to panels in *Figure 3*.

**Figure supplement 1.** Nucleotide excision repair (NER) generates long ssDNA gaps for localization of replisome components in non-replicating cells.

**Figure supplement 1—source data 1.** Source data related to panels in *Figure 3—figure supplement 1*.

the replisome that has encountered a lesion (*Belle et al., 2007*). It is unclear how such tracts are formed in non-replicating cells. We wondered whether this could be mediated via pathways involved in DNA damage repair and tolerance. Given that several repair pathways are regulated under the SOS response (*Baharoglu and Mazel, 2014*), we first assessed the induction of the response in non-replicating cells under DNA damage. For this, we measured the induction of *yfp* from an SOS-inducible promoter (P*sidA*) integrated on the *Caulobacter* chromosome at the *xyl* locus (*Badrinarayanan et al., 2015*; *Figure 3A*). We found that non-replicating cells activated the DNA damage response after MMC exposure, providing further evidence for the formation of ssDNA gaps in such conditions (*Figure 3A*). We thus asked whether the SOS response is essential for the formation of such gaps or if the activation of this response is a consequence of gap generation. Deletion of the SOS activator, *recA*, did not perturb localization of DnaN under damage. However, RecA was essential for dissociation during damage recovery as DnaN foci persisted in non-replicating cells lacking *recA* (*Figure 3B*). These observations suggest that a RecA-independent pathway is required for regulating the association of replisome components with DNA in cells that are not undergoing active DNA synthesis.

In most organisms, helix distorting lesions are recognized and excised by nucleotide excision repair (NER) (*Kisker et al., 2013*). A small proportion of the short gaps generated during this process could also be converted into longer stretches of ssDNA tracts under certain conditions, such as under high doses of DNA damage (*Cooper, 1982*; *Giannattasio et al., 2010*). This would require extensive DNA synthesis outside the active replication fork (*Figure 3—figure supplement 1A*). To test if this could be the mechanism by which replisome components associate with DNA in cells that are not replicating, we assessed the involvement of NER in orchestrating the same in *Caulobacter* swarmer cells. We observed that non-replicating cells with deletion of *uvrA* (part of the NER pathway) did not form DnaN foci under MMC or UV damage (*Figure 3B–C*, *Figure 3—figure supplement 1C*). In contrast, percentage of cells with DnaN foci in a Δ*mutL* background (deficient in mismatch repair; *Marinus, 2012*) was similar to wild type, indicating that mismatch repair did not contribute to loading of the β-clamp in non-replicating cells (*Figure 3—figure supplement 1D*).

Thus, our data suggest that lesion processing by NER alone results in the formation of ssDNA gaps on which replisome components can localize in non-replicating cells. Consistent with this, we observed lack of SSB localization in Δ*uvrA* cells both under MMC and UV damage (*Figure 3C*, *Figure 3—figure supplement 1B–C*). Furthermore, cells without NER were deficient in SOS induction (*Figure 3D*), suggesting that NER-mediated gap generation serves two functions: (a) providing ssDNA substrate for recruitment of SSB and other replisome components to these regions and (b) induction of the SOS response. Together, this facilitates ssDNA gap-filling in non-replicating *Caulobacter*.

## SOS-induced low-fidelity polymerase, DnaE2, is essential for subsequent dissociation of replisome components

As stated above, we observed that Δ*recA* cells were not deficient in DnaN recruitment to ssDNA gaps. However, given that these cells had persistent β-clamp foci, we wondered what would be the requirement for RecA or the SOS response in ssDNA gap-filling. We ruled out a role for homologous recombination in this process as our experimental setup of non-replicating swarmer cells (with a single chromosome) does not permit gap-filling by recombination, due to absence of a homologous template for repair (*Figure 1A*, bottom panel). In addition, we also conducted our damage recovery experiments in cells lacking the recombination protein RecN (*Vickridge et al., 2017*), an essential component of recombination-based repair in *Caulobacter* (*Badrinarayanan et al., 2015*). We observed similar dynamics of β-clamp foci to that seen in wild type cells in this case as well (*Figure 4—figure supplement 1A*).

Reports in *E. coli* as well as eukaryotic systems (including yeast and human cells) have suggested that ssDNA gaps generated by NER can sometimes be filled by specialized polymerases like Polκ (*Janel-Bintz et al., 2017*; *Kozmin and Jinks-Robertson, 2013*; *Sertic et al., 2018*). Given that the SOS response is activated in non-replicating cells (*Figure 3A*), it is possible that gap-filling in *Caulobacter* swarmer cells is mediated via such specialized polymerases expressed under this regulon (*Galhardo et al., 2005*). Although we were unable to generate a functional fluorescent fusion to *Caulobacter* low-fidelity polymerase DnaE2, we confirmed that DnaE2 is expressed in our experimental conditions (*Figure 4—figure supplement 1B*) and that deletion of *dnaE2* resulted in severe sensitivity of a steady-state population of cells to MMC-treatment (*Figure 2—figure supplement 1A*, *Figure 4—figure supplement 1F*). To test the involvement of DnaE2 in gap-filling, we conducted our damage recovery experiments in cells deleted for the same. Similar to Δ*recA* cells, we found that non-replicating cells lacking *dnaE2* had persistent DnaN foci during damage recovery (*Figure 4A–B*). For example, in case of wild type, 52% cells had foci after 30 min of 0.5 μg/ml MMC treatment and this number reduced to 30% 90 min post-MMC removal. In contrast, in the case of Δ*dnaE2* cells, 61% cells had foci after 30 min of damage treatment and this number remained constant even after removal of MMC from the growth media. DnaN foci in Δ*dnaE2* cells was significantly higher than wild type after 90 min of damage recovery in the case of UV damage as well, at the two doses of damage tested (*Figure 4—figure supplement 1D*).

Replisome persistence in the absence of *dnaE2* appeared to be a dose-dependent phenomenon (*Figure 4—figure supplement 1C*). At low dose of MMC treatment (0.125 μg/ml), fewer cells had DnaN foci post-DNA damage exposure (14.5% cells). The number further reduced to 9.5% during recovery in a DnaE2-independent manner. However, the percentage of cells with persistent β-clamp foci increased with increasing concentrations of damage in the absence of *dnaE2*, with minimal

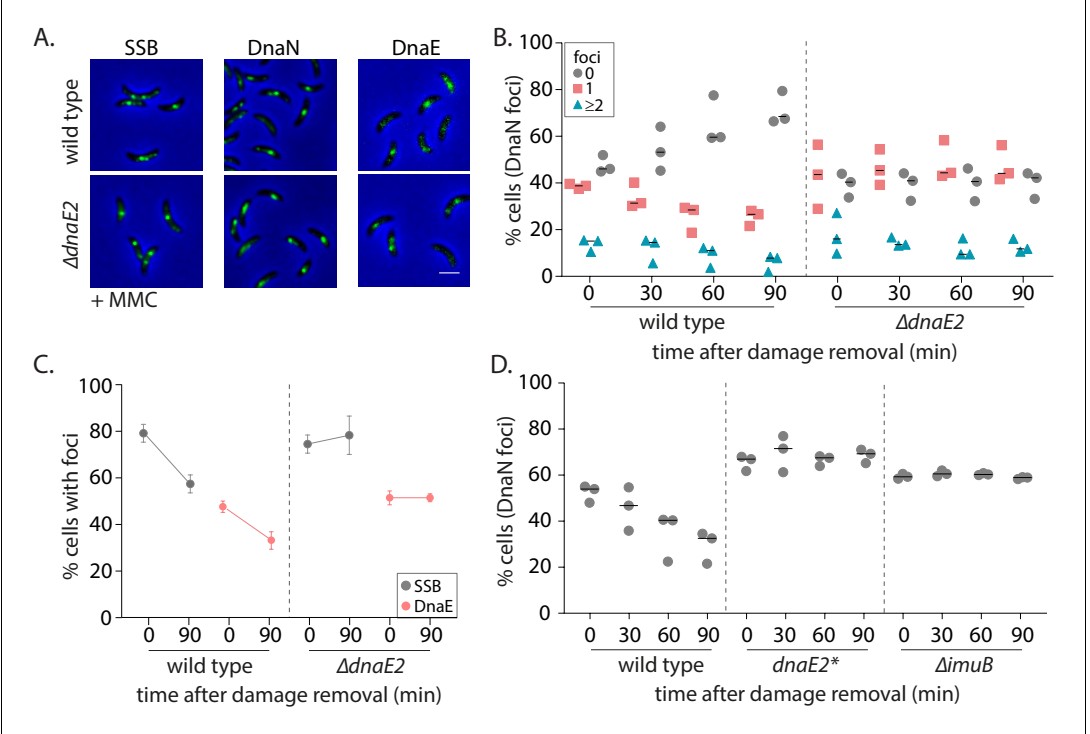

**Figure 4.** SOS-induced low-fidelity polymerase, DnaE2, is essential for subsequent dissociation of replisome components. (A) Representative images of wild type or Δ*dnaE2* swarmer cells with SSB-YFP, DnaN-YFP, or DnaE-YFP after MMC treatment. (B) Percentage wild type or Δ*dnaE2* swarmer cells with 0, 1, or ≥2 DnaN foci at 0, 30, 60, and 90 min of DNA damage recovery (n ≥ 467 cells, three independent repeats, wild type data from *Figure 2C*). (C) Percentage wild type or Δ*dnaE2* swarmer cells with SSB or DnaE foci at 0 and 90 min of DNA damage recovery (n ≥ 325 cells, mean and SD from three independent repeats). (D) Percentage wild type, *dnaE2* catalytic mutant (*dnaE2\**) or Δ*imuB* swarmer cells with DnaN foci at 0, 30, 60, and 90 min of mitomycin C (MMC) damage recovery (n ≥ 342 cells, three independent repeats, wild type data from *Figure 3B*).

The online version of this article includes the following source data and figure supplement(s) for figure 4:

**Source data 1.** Source data related to panels in *Figure 4*.
**Figure supplement 1.** SOS-induced low-fidelity polymerase, DnaE2, is essential for subsequent dissociation of replisome components.
**Figure supplement 1—source data 1.** Source data related to panels in *Figure 4—figure supplement 1*.

recovery observed at 0.5–0.75 µg/ml of MMC treatment (*Figure 4—figure supplement 1C*). The following observations in our study lend additional support to the proposed idea that a specialized polymerase is required for gap-filling across long ssDNA tracts generated by NER at higher doses of DNA damage: a. Persistence of components of PolIIIHE (DnaE and DnaN) in the absence of DnaE2. Apart from β-clamp foci, we found that the replicative polymerase, DnaE, was also unable to dissociate during damage recovery in cells lacking *dnaE2* (*Figure 4C*), suggesting that the replicative polymerase alone cannot complete synthesis across these NER-generated ssDNA tracts. Such lack of dissociation after localization was found to be the case for SSB as well, again suggesting that long ssDNA gaps persisted in the absence of DnaE2 (*Figure 4C*). b. Requirement for DnaE2-mediated synthesis. To test whether synthesis by DnaE2 contributed to gap-filling in non-replicating cells, we mutated two residues known to be essential for DnaE-mediated synthesis (*Lamers et al., 2006*; *Pritchard and McHenry, 1999*). These residues have been mutated previously in *M. smegmatis* DnaE2, where it was shown to inhibit DnaE2-dependent mutagenesis (*Warner et al., 2010*; *Figure 4—figure supplement 1E*). In the case of *Caulobacter* as well, catalytic mutant *dnaE2\** showed similar growth defects as Δ*dnaE2* under MMC damage (*Figure 4—figure supplement 1F*). In our experimental regime, we found that cells expressing catalytically inactive DnaE2 also had persistent DnaN foci during damage recovery, as seen in the case of cells lacking the specialized polymerase (*Figure 4D*).

To assess the contribution of DnaE2 in damage-induced mutagenesis, we conducted mutagenesis assays by measuring the frequency of rifampicin resistance generation in a population of cells subject to damage, either with or without recovery in non-replicating conditions. We observed that this polymerase was responsible for all damage-induced mutagenesis in our experimental regimen (*Figure 4—figure supplement 1G*). However, the genetic complexity of this experiment and the confounding effects of replication during the outgrowth period preclude us from conclusively interpreting if this mutagenesis mediated by DnaE2 occurred in non-replicating, replicating or both phases of the cell cycle.

Finally, we also assessed the requirement for the accessory protein ImuB in DnaE2 function. ImuB is an inactive Y-family polymerase and carries a β-clamp binding motif. It is thought to act as a bridge between DnaE2 and the clamp (*Warner et al., 2010*). In *Caulobacter*, it is co-operonic with *dnaE2* and is expressed in response to SOS activation (*Galhardo et al., 2005*). When we conducted our recovery experiments in cells lacking *imuB*, we observed that these cells also exhibited persistent DnaN foci, as seen for cells lacking *dnaE2* (*Figure 4D*). These results are consistent with the idea that DnaE2-mediated synthesis contributes to gap-filling and subsequent dissociation of replisome components in non-replicating cells.

## DnaE2 activity on NER-generated long ssDNA gaps enhances survival of non-replicating cells under DNA damage

Taken together, our data provide in vivo support for cross-talk between NER and specialized, low-fidelity polymerases during gap-filling in non-replicating bacteria. What could be the relevance of this in the context of damage recovery and survival of bacteria that are not actively replicating? To investigate the impact of NER-mediated DnaE2 activity in *Caulobacter* swarmer cells, we assessed the growth dynamics of these cells once released into replication-permissive conditions after damage recovery with three parameters: (a). Time to division and percentage of cells with successful division events after release in replication-permissive conditions (as a read-out for division restoration post-DNA damage clearance) (b). Cell length restoration (as a read-out for SOS deactivation following DNA damage clearance). (c). Cell survival measured via viable cell count assays.

To measure division restoration, we released replication-blocked swarmer cells into media containing IPTG (to allow for replication initiation via induction of *dnaA*) either immediately after damage treatment or after 90 min of damage recovery. We followed single cells via time-lapse imaging to assess the time taken to first division after replication initiation (*Figure 5A–B*). Control cells without damage treatment and with or without an additional 90 min arrest in swarmer stage were able to robustly resume cell growth and division, with >94% cells undergoing their first division within 240 min of release into replication-permissive conditions. Based on this, we followed cell division dynamics for cells treated with damage during this time window, wherein control cells (without damage) were successfully able to restore cell division. In MMC-treated conditions, we found that cells released into replication-permissive conditions immediately after damage treatment did not recover efficiently, with only 5% cells undergoing their first division within 240 min (*Figure 5C*). In contrast, wild type cells that were provided time for damage recovery before reinitiating replication showed restoration of cell division in the same time period, with 30% cells undergoing at least one division and 9% cells undergoing ≥2 divisions within 240 min (*Figure 5B–C*). These recovery dynamics were dependent on DnaE2 as only 7% cells lacking *dnaE2* underwent divisions even when they were provided the same time duration as wild type for damage recovery before replication reinitiation (*Figure 5B–C*). Thus, DnaE2-mediated gap-filling provided significant survival advantage to non-replicating cells as measured by their ability to robustly restore cell cycle progression and cell division.

To further assess the consequence of gap-filling, we measured the cell length distributions for cells released into replication-permissive conditions with or without 90 min of DNA damage recovery (*Figure 5—figure supplement 1A*). Continued cell length elongation would be reflective of a continued division block, a hallmark of the SOS response. On the other hand, cell length restoration would be expected only for those cells where damage has been repaired (*Raghunathan et al., 2020*). We found that cells that did not face damage (with or without *dnaE2*) had a median cell length of 4.6 μm after 90 min incubation in swarmer conditions. At 240 min after reinitiation of replication, the cell length distribution was restored close to a wild type-like pattern (control) with the median cell length dropping to 2.9 μm (*Figure 5—figure supplement 1A*, 'no damage'). Length restoration was also observed in wild type cells able to engage in DnaE2-mediated gap-filling in the 90 min recovery

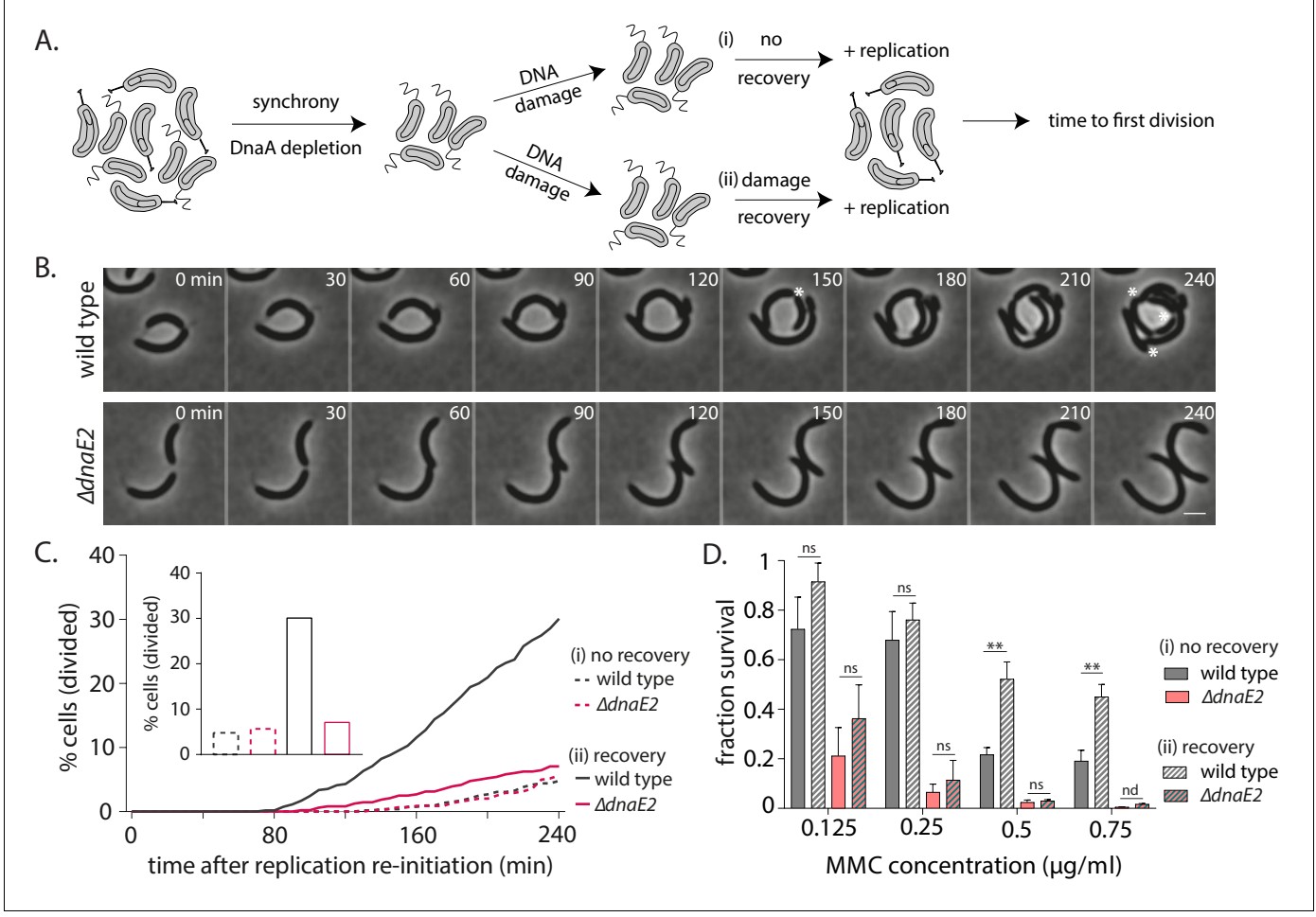

**Figure 5.** DnaE2 activity on nucleotide excision repair (NER)-generated long single-stranded DNA (ssDNA) gaps enhances survival of non-replicating cells under DNA damage. (A) Schematic of experimental setup used to assess the impact of lesion repair/ tolerance in non-replicating cells. After mitomycin C (MMC) treatment for 30 min, cells were either released into replication-permissive media (i: no recovery) or allowed to grow for 90 min without damage and then released into replication-permissive media (ii: damage recovery). Cells were followed via time-lapse microscopy and time to division was estimated. Control cells were taken through the same growth regimes; however, no damage is added to the culture. (B) Representative time-lapse montage of wild type or ΔdnaE2 cells in replication-permissive media after DNA damage recovery. Cell divisions are marked with white asterisk. In the panel shown here, three divisions were scored in wild type, while none were observed in ΔdnaE2 cells. (C) Percentage cell division over time after replication reinitiation for wild type and ΔdnaE2 cells either without (i: no recovery) or with (ii: recovery) damage recovery time in replication-blocked conditions (n ≥ 368 cells). Inset: Percentage cells divided at 240 min in each of these conditions is summarized. (D) Survival of wild type and ΔdnaE2 cells either without (i: no recovery) or with (ii: recovery) damage recovery time in replication-blocked conditions was measured via estimation of viable cell count (three independent repeats). Fraction survival was calculated by normalizing viable cell count under DNA damage to that without DNA damage (mean with SD from three independent experiments).

The online version of this article includes the following source data and figure supplement(s) for figure 5:

**Source data 1.** Source data related to panels in *Figure 5*.

**Figure supplement 1.** DnaE2 activity on nucleotide excision repair (NER)-generated long single-stranded DNA (ssDNA) gaps enhances survival of non-replicating cells under DNA damage.

**Figure supplement 1—source data 1.** Source data related to panels in *Figure 5—figure supplement 1*.

window (*Figure 5—figure supplement 1A*, '+ damage, recovery, wild type'). This restoration in cell length was dependent on the time provided for damage recovery as well as presence of DnaE2. In the absence of recovery or *dnaE2*, cells continued to elongate after release into IPTG-containing media (*Figure 5—figure supplement 1A*, '+damage, no recovery' and '+damage, recovery, ΔdnaE2').

To lend support to these cell biological observations, we modified our recovery setup to measure viable cell counts (*Figure 5—figure supplement 1B*). For this, we assessed the 'fraction survival' as defined by the viable cell count obtained for cultures with damage treatment normalized to the viable cell count for cultures without damage treatment. We observed that wild type cells that were released into replication-permissive conditions without the 90 min window of damage recovery were significantly compromised in growth, with fraction survival reducing to 0.19 at higher doses of damage in the absence of recovery. On the other hand, in case of cells grown with the possibility of undergoing 90 min of damage recovery, the fraction survival increased to 0.45 at the highest dose of damage used (*Figure 5D*). We then asked whether the survival advantage observed during replication-independent damage recovery required DnaE2 action. Consistent with a dose-dependent effect on replisome persistence in the absence of DnaE2, we also observed that DnaE2 had a significant impact on the replication-independent survival advantage at higher doses of DNA damage. We found that cells deleted for *dnaE2* were severely compromised for survival at all doses of damage used (*Figure 5D*). However, at higher doses of damage, cells lacking *dnaE2* had similar reduction in viable cell counts whether or not they were given a 90 min window of recovery. For example, after treatment with 0.5 µg/ml of MMC, only 0.01 fraction survival was observed for cells lacking *dnaE2* (with or without damage recovery). On the other hand, wild type cells which had 90 min of damage recovery showed a fraction survival of 0.45 (*Figure 5D*). Thus, there was a significant component of enhanced survival in cells that could undergo repair in non-replicating conditions and this survival advantage was dependent on DnaE2.

In summary, our cell biological and genetic read-outs suggest that DnaE2-mediated gap-filling enables cell cycle restoration and cell division licensing when non-replicating cells are allowed to reinitiate DNA replication. In the absence of such recovery (either *dnaE2* deletion or cells grown without time for recovery), cell division is compromised and cells continue to elongate, a hallmark of persistent DNA damage and hence continuously active SOS response. The impact of delayed cell division and subsequent cell length elongation is directly observed when viable cell count of the population is measured, with a dose-dependent effect on survival in cells compromised for recovery due to deletion of *dnaE2*.

## Discussion

DNA lesion repair and tolerance have been well studied in a replication-centric paradigm (*Gabbai et al., 2014*; *Indiani et al., 2005*; *Marians, 2018*). Characterization of error-prone polymerases in *E. coli* has informed us about mechanisms of tolerance that could occur at the replication fork or behind it, in gaps generated due to replisome skipping over the lesion, followed by repriming downstream of it (*Chang et al., 2019*; *Gabbai et al., 2014*; *Indiani et al., 2005*). However, DNA damage is a universal event that can occur across all stages of the cell cycle, including in non-replicating conditions. This can have effects on transcription and could also perturb replication progression upon reinitiation (*Jeiranian et al., 2013*; *Lang and Merrikh, 2018*; *Rudolph et al., 2007*). For example, bacteria such as *Caulobacter* have distinct cell cycle phases including a non-replicating swarmer state, with a single copy of its chromosome. Hence, it is imperative that DNA damage is repaired efficiently even in these conditions. Here, we provide in vivo evidence for NER-coupled DnaE2 function that is active in non-replicating bacteria. This study complements a growing body of work that supports the possibility of low-fidelity polymerase-mediated synthesis (including mutagenesis) in replication-independent conditions (such as in stationary phase cells) across domains of life (*Bull et al., 2001*; *Corzett et al., 2013*; *Janel-Bintz et al., 2017*; *Sung et al., 2003*; *Yeiser et al., 2002*) and underscores the need to reconsider function of such polymerases outside canonical, isolated roles of lesion bypass during replication.

### DNA damage repair and tolerance in non-replicating cells: requirement for DnaE2

Here, we develop a system to specifically assess mechanisms of damage repair and tolerance employed in cells that are not undergoing active DNA synthesis. Using replication initiation-inhibited *Caulobacter* swarmer cells, we show that lesions can be dealt with in two main steps: a. damage processing by NER to reveal SSB-bound long ssDNA gaps and b. gap-filling by SOS-induced specialized polymerase, DnaE2. Due to absence of a second copy of the chromosome in our assay (all cells

are non-replicating and have a single chromosome), role for homologous recombination in this process is unlikely. Hence, our observations are consistent with a scenario where the low-fidelity polymerase alone is sufficient to synthesize across these long ssDNA gaps generated by NER action. Why is there a need for a specialized polymerase during gap-filling of NER-generated substrates? We explore two possible scenarios here:

(1) Conventionally NER is thought to generate gaps of approximately 12 nucleotides during lesion repair, which can be gap-filled by DNA PolI (*Kisker et al., 2013*). However, localization of SSB in our experiments suggests the presence of gaps > 30 nucleotides, enabling SSB tetramerization and binding (*Bell et al., 2015*; *Lohman and Ferrari, 1994*). How are longer ssDNA tracts generated? Previous reports in *E. coli* as well as yeast and human cells have implicated a role for exonuclease activity in generating longer ssDNA tracts on some NER substrates. In these studies, it was proposed that such activity would occur on problematic intermediates generated during NER activity, including closely-spaced opposing lesions that are generated under high doses of DNA damage (*Janel-Bintz et al., 2017*; *Kozmin and Jinks-Robertson, 2013*; *Sertic et al., 2018*). Indeed, our observations on lack of dissociation of replicative polymerase (PolIII) in the absence of DnaE2 as well as the dose-dependent impact on cell survival would both be consistent with a speculative model where NER-mediated excision results in the production of lesion-containing ssDNA that requires synthesis by a specialized polymerase.

(2) It is equally plausible that DnaE2 contributes to gap-filling independent of the presence or absence of a DNA lesion. Earlier studies in *E. coli* indicated that a minor component of NER-mediated removal of UV lesions can result in long ssDNA gaps that are gap-filled in a process referred to a long patch excision repair (LPER) (*Cooper, 1982*). Though the molecular players of this long patch synthesis are unidentified, this process did not result in detectable mutations. Furthermore, studies in yeast have implicated a role for exonuclease activity (via Exo1) in generating long gaps during a subset of NER events which drive checkpoint activation and are eventually filled via long patch repair synthesis (*Giannattasio et al., 2010*). DnaE2 could function similarly in gap-filling on these long ssDNA gaps formed as a consequence extensive NER in the context of severe DNA damage. Gap-filling activity has been suggested previously for specialized polymerases such as eukaryotic Polκ (*Ogi and Lehmann, 2006*). In addition, recent studies on post-replicative gap-filling have proposed a scenario where long patches requiring synthesis are accessed by both replicative and TLS polymerases (PolIV and PolV) in *E. coli* (*Isogawa et al., 2018*). In the case of non-replicating *Caulobacter* cells, it is possible that DnaE2 can access the β-clamp and hence participate in such gap-filling, given the observed increase in DnaE2 levels via SOS induction.

A limitation of our current study is that we do not observe all NER events, a significant proportion of which could be mediated via gap-filling by PolI on short ssDNA stretches. The relative contribution of these two arms of NER (long vs. short patch repair) could vary with increasing doses of damage and subsequently impact the requirement for DnaE2 action in gap-filling. Unfortunately, using our mutagenesis assays (measuring generation of rifampicin resistant mutations during damage), we were unable to satisfactorily disentangle the individual contributions of DnaE2-mediated mutagenesis in non-replicating vs. replicating conditions (*Figure 4—figure supplement 1G*). Hence, we cannot reliably distinguish between the 'gap-filling alone' or 'gap-filling associated with lesion bypass' activities of this polymerase in our present study. It must be noted though, that a role for DnaE2 in gap-filling alone has not been reported before. In addition, unlike *E. coli,* it is the only polymerase implicated in TLS-associated functions (mutagenesis) in the bacteria that encode it. However, irrespective of the specific nature of DnaE2 activity, our work underscores a novel and necessary function for this highly conserved specialized polymerase in conjunction with NER in replication-independent conditions.

## Long ssDNA gaps generated by NER serve two functions

Previous studies in *E. coli* have found that NER activity in GGR is dependent on the activation of the SOS response (*Crowley and Hanawalt, 1998*). In contrast, our results suggest that NER functions upstream of the SOS response in non-replicating *Caulobacter* in the context of long patch repair. Although *uvr* genes are SOS-induced even in *Caulobacter* (*da Rocha et al., 2008*), it is possible that basal levels of Uvr proteins are sufficient to carry out damage scanning and subsequent lesion processing. Indeed, in *E. coli,* basal UvrA levels are variable, but range from 9 to 43 copies in minimal

media and more than 120 copies in rich media (*Ghodke et al., 2020*). Long ssDNA gaps generated by NER serve two purposes:

(a) Activation of the SOS response for specialized polymerase expression; it is likely that in case of *Caulobacter*, RecA is essential only for turning on the SOS regulon as DnaE2-mediated synthesis has been previously shown to function independent of RecA (*Alves et al., 2017*; *Galhardo et al., 2005*), unlike *E. coli* PolV (*Goodman, 2014*; *Nohmi et al., 1988*).

(b) Providing substrate for SSB and PolIIIHE localization and specialized polymerase-mediated gap-filling. SSB localization on ssDNA could further facilitate recruitment and loading of the PolIIIHE. While PolIII activity could directly contribute to gap-filling (*Isogawa et al., 2018*; *Sedgwick and Bridges, 1974*; *Soubry et al., 2019*), it is also likely that it is the loading of the β-clamp that is essential for DnaE2 activity (*Bunting et al., 2003*; *Chang et al., 2019*; *Fujii and Fuchs, 2004*; *Wagner et al., 2009*). Additionally, recent studies have highlighted a role for SSB as well in enriching the local pool of PolIV at a lesion, thus enabling polymerase switching (*Chang et al., 2020*). The lack of a significant percentage of cells with multiple replisome foci under damage would suggest that only a limited number of long ssDNA gaps are generated per cell or that some repair/replisome component involved in gap processing or gap-filling is limiting.

It would be interesting now to ask how these additional components (such as ImuB and other accessory components to DnaE2) contribute to the regulation of the 'specialized replisome' outside the realms of active replication and whether the properties of the ssDNA gaps generated vary under damaging conditions that result in different types of lesions (CPDs in UV vs. monoadducts and cross-links in MMC) (*Bargonetti et al., 2010*; *Chatterjee and Walker, 2017*; *Mitchell and Nairn, 1989*). Indeed, although discussed in the context of non-replicating cells, it is plausible that, under high doses of damage, this mechanism can occur spatially and temporally disconnected from the active replication fork in replicating cells as well, in support of observations in *E. coli* that have reported localization of PolIIIHE and specialized polymerases away from the active replication fork (*Henrikus et al., 2018*; *Soubry et al., 2019*).

## Relevance of NER-mediated specialized polymerase activity in non-replicating cells

Our study provides comprehensive insights into a mechanism of lesion repair and gap-filling in non-replicating bacteria, which relies on a coordinated action between NER and low-fidelity polymerases. Our data suggests a method through which an error-prone polymerase, DnaE2, functions beyond replication forks, impinging on its implications in growth and survival of non-replicating cells. The experimental system in this study provides a novel tool to investigate these mechanisms as well as additional players further and assess impacts of lesion repair and tolerance in replication-independent, but metabolically active conditions, where damage to DNA via molecules including ROS is possible (*Gray et al., 2019*; *Manina and McKinney, 2013*) (such as *Caulobacter* cells in 'swarmer' state or other cells outside S phase of cell cycle).

The relevance of the process described here is highlighted by the survival advantage it confers to non-replicating cells. It is possible that NER-coupled DnaE2-mediated gap-filling helps avoid the problems associated with persistent ssDNA gaps (due to extensive NER activity itself) or DNA damage on the chromosome (*Jeiranian et al., 2013*; *Murli et al., 2000*; *Rudolph et al., 2007*). For example, recent study in human cells showed that coordinated action of NER along with Y-family polymerase, Polκ, and exonuclease, Exo1, was crucial for gap-filling and prevention of UV-induced double-stranded breaks in non-S phase cells (*Sertic et al., 2018*). Such a role for specialized polymerases in gap-filling has also been observed in case of yeast cells (*Kozmin and Jinks-Robertson, 2013*; *Sertic et al., 2011*). More generally, our work highlights the possibility of coordinated activity of repair and tolerance pathways canonically studied as functioning independently. The universality of the NER-mediated error-prone polymerase function described here is underscored by its functionality in a diverse range of model systems, from bacteria to yeast and human cells (*Janel-Bintz et al., 2017*; *Kozmin and Jinks-Robertson, 2013*; *Sertic et al., 2018*), independent of the type or family of error-prone polymerase (DnaE2 in *Caulobacter* vs. PolIV/ PolV in *E. coli*) employed during gap-filling.

# Materials and methods

## Key resources table

| Reagent type (species) or resource | Designation | Source or reference | Identifiers | Additional information |
|---|---|---|---|---|
| Strain, strain background *Caulobacter crescentus* NA1000 | *Caulobacter crescentus* NA1000 strains | PMID:334726 This study | | *Supplementary file 1* |
| Recombinant DNA reagent | Plasmids | This study | | *Supplementary file 2* |
| Sequence based reagents | Oligos | This study | | *Supplementary file 3* |
| Antibody | Anti-Flag (mouse monoclonal) | Sigma-Aldrich | F1804 (RRID:AB_262044) | Western blot (1:2000) |
| Antibody | Anti-mouse IgG, HRP-linked antibody | Cell Signaling Technology | 7076S (RRID:AB_330924) | Western blot (1:5000) |
| Commercial assay, kit | SuperSignal West Pico Plus Chemiluminescent Substrate | Thermo Scientific | 34577 | Western blot |
| Chemical compound, drug | Mitomycin C (MMC) | AG Scientific | M-2715 | DNA damaging agent |
| Commercial assay, kit | SYTOX Green Nucleic Acid Stain | Thermo Fisher Scientific | S7020 | Flow cytometry |
| Chemical compound, drug | Percoll | GE Healthcare | 17-0891-01 | Synchrony |
| Software, algorithm | GraphPad Prism 8 | GraphPad Software | RRID:SCR_002798 | Analysis |
| Software, algorithm | Fiji (ImageJ) | *Schindelin et al., 2012* | RRID:SCR_002285 | Analysis |
| Software, algorithm | MATLAB R2020a | MathWorks | RRID:SCR_001622 | Analysis |
| Software, algorithm | Oufti | *Paintdakhi et al., 2016* | RRID:SCR_016244 | Analysis |
| Software, algorithm | MicrobeTracker | *Sliusarenko et al., 2011* | RRID:SCR_015939 | Analysis |

## Bacterial strains and growth conditions

Bacterial strains, plasmids, and primers used in the study are listed in Key Resources Table (and *Supplementary files 1–3*). Construction of plasmids and strains is also detailed in respective supplementary files. Transductions were performed using ΦCR30 (*Ely, 1991*). *Caulobacter crescentus* cultures were grown at 30°C in PYE media (0.2% peptone, 0.1% yeast extract and 0.06% $MgSO_4$) supplemented with appropriate concentrations of antibiotics, as required. While growing strains carrying *dnaA* under an IPTG-inducible promoter, liquid media were supplemented with 0.5 mM IPTG and solid media with 1 mM IPTG. Microscopy experiments were performed in minimal media containing 1X M2 salts (0.087% $Na_2HPO_4$, 0.53% $KH_2PO_4$, 0.05% $NH_4Cl$) supplemented with 1% PYE, 0.2% glucose, 0.01 mM $FeSO_4$, and 0.01 mM $CaCl_2$.

Non-replicating swarmer cells were isolated using synchrony protocols described previously (*Badrinarayanan et al., 2015*; *Chimthanawala and Badrinarayanan, 2019*). Briefly, cells were grown overnight in minimal media supplemented with IPTG. Cultures in log-phase were depleted for DnaA via washing off IPTG and allowing cells to grow in IPTG (-) conditions for one generation (~130 min). Following this, cultures were synchronized and $OD_{600}$ of resulting swarmer cells was adjusted to 0.1, prior to treatment with DNA damage. In case of MMC damage, appropriate volume of 0.5 mg/ml MMC (AG Scientific, #M-2715) stock (prepared by resuspending in sterile water) was added into the culture and incubated at 30°C for 30 min. Damage was washed off by pelleting down cells at 8000 rpm for 4 min and resuspending in fresh media. For UV damage, cultures were transferred to a 90 mm petri plate and exposed to specific energy settings in a UV Stratalinker 1800 (STRATAGENE). During recovery (after UV and MMC damage), cells were incubated for 90 min at 30°C and 200 rpm. For strains expressing *SSB-YFP*, *SSB-GFP*, or *dnaN-YFP* under $P_{xyl}$, 0.3% xylose was added 1.5 hr prior to imaging. Replication reinitiation after damage recovery was achieved by inducing cultures

with 0.5 mM IPTG. DNA damage treatment used was either 0.5 µg/ml MMC (30 min) or 75 J/m$^2$ UV for all experiments, unless otherwise specified.

For flow cytometry analysis, 300 µl of cultures were fixed in 700 µl of 70% chilled ethanol and stored at 4°C until further processing. These samples were treated with 2 µg/ml RNaseA in 50 mM sodium citrate for 4 hr at 50°C. DNA was stained with Sytox green nucleic acid stain (5 mM solution in DMSO from Thermo Fisher Scientific) and analyzed using a BD Accuri flow cytometer.

## Fluorescence microscopy and image analysis

For time course imaging, 1 ml aliquots of cultures were taken at specified time points, pelleted, and resuspended in 100 µl of growth medium. Images were taken without damage treatment (no damage control), after 30 min of damage treatment (+ damage) and again at 0, 30, 60, and 90 min after removal of DNA damage (recovery). Control cells were grown under same treatment regime, but no damaging agent was added to growth media. 2 µl of cell suspension was spotted on 1% agarose pads (prepared in minimal medium) and imaged. For time-lapse imaging, 2 µl cell suspension was spotted on 1.5% GTG agarose (prepared in minimal medium), grown inside an OkoLab incubation chamber maintained at 30°C and imaged at specific intervals for the indicated period of time. For cell division tracking after replication reinitiation, cells were grown on 1.5% GTG agarose prepared in growth medium with 1 mM IPTG.

Microscopy was performed on a wide-field epifluorescence microscope (Eclipse Ti-2E, Nikon) with a 63X oil immersion objective (plan apochromat objective with NA 1.41) and illumination from pE4000 light source (CoolLED). The microscope was equipped with a motorized XY stage and focus was maintained using an infrared-based Perfect Focusing System (Nikon). Image acquisitions were done with Hamamatsu Orca Flash 4.0 camera using NIS-elements software (version 5.1). For excitation at 460 nm, exposure time was set to 300 ms; at 490 nm, exposure time used was 400 ms; and for 550 nm, exposure time of 300 ms was used. Images were analyzed using ImageJ as well as Microbetracker or Oufti in MatLab (*Paintdakhi et al., 2016*; *Sliusarenko et al., 2011*). Values for random positions within each cell and relative position of replisome foci were generated using the following custom-written MatLab scripts.

```
load(''); %mesh_file from oufti
%% extract data for position of spots and cell ids from spots file;
k = 1;
for i = 1:length(cell_List{1,1})
   if isfield(cell_List{1,1}{1,i}, 'spots')==1
     for j = 1:length(cell_List{1,1}{1,i}.spots.l)
        cell_position(k,1)= cell_List{1,1}{1,i}.spots.l(1,j);
        cellids(k,1) = i;
        k=k+1;
     end
   else
     continue
   end
end
%% calculate cell lengths from mesh file
for i = 1:length(cellids)
   var = cellids(i);
   cell_length(i,1)= length(cellList.meshData{1, 1}{1, var}.mesh);
end
%% generate random floating point numbers
for i = 1:length(cell_position)
   if cell_length(i,1)==0
     random_num(i,1)=0;
   else
      random_num(i,1) = rand(1)+ randi(cell_length(i,1)-1); %randomly generated
numbers
   end
```

```
end
%% distance between random variable to dnaN
for i = 1:length(random_num)
   dist_pix(i,1) = abs(random_num(i,1)-cell_position(i,1)); %in pixels
end
%in microns
dist_micr= dist_pix*0.108;
```

Graphs were generated in GraphPad Prism 8. Statistical analysis was performed in GraphPad Prism 8. Exact p-values are summarized in *Supplementary file 4*.

## Survival assay

For calculating viability of an asynchronous steady-state population under DNA damage (*Figure 2—figure supplement 1A and D*), *Caulobacter* cultures were grown in PYE with 0.5 mM IPTG to $OD_{600}$ of 0.3. For assessing survival under MMC, serial dilutions were made in 10-fold increments and 6 µl of each dilution ($10^{-1}$ to $10^{-8}$) was spotted on PYE agar containing 1 mM IPTG and appropriate amounts of MMC. For assessing survival under UV, similar serial dilutions were made, spotted on PYE agar containing 1 mM IPTG and exposed to appropriate doses of UV in a UV Stratalinker 1800 (STRATAGENE). Growth was quantified by multiplying dilution factor of the last visible spot with number of colonies on the last spot. Percentage survival for each strain was calculated by normalizing growth of that specific strain treated with different doses of DNA damage to that in media without DNA damage.

For assessing survival of non-replicating cells under MMC (*Figure 5D*), non-replicating swarmer cells (10 ml, $OD_{600}$ 0.1) were treated with different concentrations of DNA damage for 30 min. After washing off damage, these replication-blocked cells were taken through either 'damage recovery' (90 min recovery) or 'no recovery' regime. Cells from both regimes were serially diluted, plated on PYE agar containing 1 mM IPTG, and colony counts were estimated after 48 hr. Fraction survival was calculated by normalizing viable cell count of MMC-treated cells to viable cell count without DNA damage treatment. Refer *Figure 5—figure supplement 1B* for schematic of the experimental setup.

## Rifampicin resistance assay

Non-replicating swarmer cells (10 ml, $OD_{600}$ 0.1) were grown in 'no recovery' or 'damage recovery' conditions (as described above for survival experiments; *Figure 5—figure supplement 1B*). At the end of each experimental treatment, cultures were spun down, resuspended in 10 ml PYE containing 0.5 mM IPTG, and grown at 30°C overnight. These cultures were plated on PYE agar containing 1 mM IPTG and 100 µg/ml Rifampicin. Rifampicin-resistant colonies were counted 48 hr after plating, and mutation frequencies were calculated by normalizing to viable cell count for that specific culture.

## Western blotting

At specific time points of the experiment, 1.5 ml aliquots of 0.1 $OD_{600}$ cultures were pelleted down at 10000 rpm for 5 min, pellets were snap frozen in liquid nitrogen and stored at −80°C. Pellets were resuspended in SDS sample buffer, and boiled at 95°C for 10 min. Equal amounts of lysates were loaded on 6% SDS-PAGE gel, resolved at 100 V and transferred to polyvinylidene fluoride (PVDF) membrane (BIO-RAD, #1620177) in a wet electroblotting system. Non-specific binding to the membrane was blocked with 5% Blotting-Grade Blocker (BIO-RAD, #170–6404), followed by probing with 1:2000 dilution of monoclonal anti-flag antibody (Sigma, #F1804, RRID:AB_262044) and 1:5000 dilution of HRP-linked anti-mouse secondary antibody (Cell Signaling Technology, #7076S, RRID:AB_330924). The blots were visualized after incubation with SuperSignal West PICO PLUS Chemiluminescent Substrate (Thermo Scientific, #34577) using an iBright FL1000 imager (ThermoFisher Scientific).

## Acknowledgements

The authors acknowledge assistance from Prachi Shinde and Dr. Ramya Rajagopalan for preliminary experiments and NCBS Central Imaging and Flow Facility (CIFF) for flow cytometry usage. The

authors thank Dr. Rodrigo Reyes-Lamothe, Dr. Sabari Tirupathy, Dr. Stephan Uphoff, Dr. Tung Le, and members of the AB lab for helpful discussions and feedback on the manuscript. The authors are grateful for the constructive feedback and suggestions from the reviewers and editors. This work was supported by fellowships from DBT (DBT-RA), DST-SERB (PDF/2018/001164) (AMJ), and CSIR (SD) as well as grants from DBT-IYBA (BT/12/IYBA/2019/10), HFSP CDA (00051/2017 C), and intramural funding from NCBS-TIFR (Department of Atomic Energy, Government of India, under project no. 12 R and D-TFR-5.04–0800) (AB).

## Additional information

### Funding

| Funder | Grant reference number | Author |
| --- | --- | --- |
| Human Frontier Science Program | 00051/ 2017-C | Anjana Badrinarayanan |
| Department of Atomic Energy, Government of India | 12-R&D-TFR-5.04-0800 | Anjana Badrinarayanan |
| Department of Science and Technology, Ministry of Science and Technology | PDF/2018/001164 | Asha Mary Joseph |
| Department of Biotechnology , Ministry of Science and Technology | IYBA(BT/12/IYBA/2019/10) | Anjana Badrinarayanan |
| CSIR | | Saheli Daw |

The funders had no role in study design, data collection and interpretation, or the decision to submit the work for publication.

### Author contributions

Asha Mary Joseph, Conceptualization, Resources, Data curation, Formal analysis, Funding acquisition, Validation, Investigation, Visualization, Methodology, Writing - original draft, Writing - review and editing; Saheli Daw, Resources, Data curation, Formal analysis, Validation, Investigation, Visualization, Writing - review and editing; Ismath Sadhir, Resources, Formal analysis, Validation, Writing - review and editing; Anjana Badrinarayanan, Conceptualization, Data curation, Formal analysis, Supervision, Funding acquisition, Validation, Investigation, Visualization, Methodology, Writing - original draft, Project administration, Writing - review and editing

### Author ORCIDs

Asha Mary Joseph https://orcid.org/0000-0002-0465-9799
Anjana Badrinarayanan https://orcid.org/0000-0001-5520-2134

### Decision letter and Author response

Decision letter https://doi.org/10.7554/eLife.67552.sa1
Author response https://doi.org/10.7554/eLife.67552.sa2

## Additional files

### Supplementary files

- Supplementary file 1. Table for strains used in the study and strain construction details.
- Supplementary file 2. Table for plasmids used in the study and cloning details.
- Supplementary file 3. Table for oligonucleotides used in the study.
- Supplementary file 4. Summary of p-values for statistical tests performed in the study.
- Transparent reporting form

## Data availability

Data analysed during this study are included in the manuscript. Numerical data files (source data files) have been provided for Figure 1—figure supplement1, Figure 2–5 and corresponding figure supplements.

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
