## [Decision Letter]

**Acceptance summary:**

In this paper, the authors showed that DnaE2, a bacterial translesion synthesis DNA polymerase participates in long patch excision repair, a sub-pathway of nucleotide excision repair. This paper will be of substantial interest to those in the fields of DNA repair and mutagenesis. The authors employ a novel approach to identify a new function for DnaE2 in bacteria linked to cell survival, thus adding to a growing list of activities for DNA polymerases outside the context of the DNA replication fork. While the study relies on indirect observations, significant novel insight is gained and the conclusions are consistent with the data.

**Decision letter after peer review:**

[Editors’ note: the authors submitted for reconsideration following the decision after peer review. What follows is the decision letter after the first round of review.]

Thank you for submitting your work entitled "Coordination of NER with replication-independent translesion synthesis is essential for bacterial DNA damage survival" for consideration by *eLife*. Your article has been reviewed by 3 peer reviewers, and the evaluation has been overseen by a Reviewing Editor and a Senior Editor. The following individuals involved in review of your submission have agreed to reveal their identity: Robert Fuchs (Reviewer #1); Andrew Robinson (Reviewer #2).

Our decision has been reached after consultation between the reviewers. Based on these discussions and the individual reviews below, we regret to inform you that your work will not be considered further for publication in *eLife*.

This work proposes an intriguing concept that TLS-mediated synthesis takes place on NER-generated gaps in the absence of DNA replication. After discussing the manuscript, the reviewers and the reviewing editor agreed that while the links between NER and DnaE2 are interesting, the work falls short of proving that DnaE2 is participating in TLS and that it more likely has a role in gap resolution that does not depend on lesion bypass. All three reviewers have clear concerns about the level of mechanistic insight provided by the manuscript. The individual reviews provides some suggestions of how such an insight may be achieved, but that would require a significant amount of additional work.

*Reviewer #1:*

My overall comment is that the data do support neither the title nor the abstract of the paper. In the title "bacterial DNA damage survival" is mentioned while no damage survival data are provided. In the abstract, the authors say they provide evidence for "replication-independent TLS". At the best their data show that a TLS polymerase, namely DnaE2, participate into NER gap-filling which does not mean that DnaE2 performs a lesion bypass. Indeed, a gap generated during NER does in principle not contain a lesion except in the rare situations of closely spaced lesions. If this were indeed the case the authors should observe a quadratic dose response. The involvement of DnaE2 during NER does not mean that DnaE2 is performing Translesion synthesis (TLS). There are more and more situations describing the involvement of TLS polymerases in transactions that do not involve lesion bypass per se. For example, the involvement of Pol kappa has been described many years ago during NER-gap filling in yeast and in human cells without bona fide lesion bypass (see ref below). To assess genuine lesion bypass the authors would need to monitor induced mutagenesis that is the signature of TLS.

Substantive concerns:

1. Proper description of the level of damage induced by MMC and UV irradiation should be provided. At the best I can tell the authors only used one dose of MMC and one dose of UV. There needs to be experiments showing the physiological effect of MMC treatment or of UV irradiation such as survival (as claimed in the title) or mutagenesis.

2. All data are based on fluorescence imaging. Other methodologies (genetics…..) should be implemented to reinforce the study.

Ogi T, Lehmann AR. The Y-family DNA polymerase kappa (pol kappa) functions in mammalian nucleotide-excision repair. Nat Cell Biol. 2006; 8: 640-642. https://doi.org/10.1038/ncb1417 PMID: 16738703

Lehmann AR. New functions for Y family polymerases. Mol Cell. 2006; 24: 493-495. https://doi.org/10.1016/j.molcel.2006.10.021 PMID: 17188030

*Reviewer #2:*

Joseph et al. report a single-molecule fluorescence microscopy study that reveals interesting new links between nucleotide excision repair and the translesion synthesis DNA polymerase DnaE2 in non-replicating *Caulobacter crescentus* cells. By analysing non-replicating cells, the authors are able to examine replication/repair activities that occur outside of the context of the replication fork. The data provide strong support for a functional link between NER and TLS occurring under these conditions. The mechanism they propose is reasonable, however I think that an alternative mechanism that involves homologous recombination would fit their data equally well. Decoupling DnaE2 expression from the RecA*-mediated SOS response should allow the authors to distinguish between the two mechanisms.

1. It seems possible to me that the gaps created by the Uvr proteins might be repaired via homologous recombination, as opposed to gap-filling. The recruitment of SSB, HolB, DnaN, and DnaE to the repair intermediates, and the strong dependence on *recA* would be consistent with this idea. The requirement for DnaE2 could be consistent with a role in recombination, such as D-loop extension. As the authors point out, the footprint of SSB is larger than the typical gaps produced by UvrABC, so SSB foci should not form on these short gaps. SSB foci might instead form if the initial gaps are enlarged in preparation for homologous recombination (perhaps through the actions of RecQ and RecJ, as has been reported in *E. coli*). The authors also report robust induction of the SOS response. This requires the formation of RecA* nucleoprotein filaments, such as those formed in preparation for homologous recombination. It might be possible for RecA* to form on short gaps produced by UvrABC, but it seems more likely (and consistent with the formation of SSB foci) that RecA* is forming on gaps that have been enlarged for homologous recombination. The observation that *recA* is not required for the formation of DnaN foci is consistent with the notion that clamps are loaded at gaps created by UvrABC. It remains formally possible, however, that clamps are loaded at recombination intermediates in the presence of RecA, and at SSB-coated gaps in its absence. Both scenarios would lead to focus formation.

2. Decoupling the production of DnaE2 from the formation of RecA* should allow the authors to distinguish between their proposed gap-filling mechanism and a homologous recombination mechanism. They could do this by expressing DnaE2 from a plasmid, or by introducing a *lexA* null mutation to make the cells SOS constitutive. Each approach has advantages and disadvantages, so ideally both would be tested. If a gap-filling mechanism is at play, the cells should no longer require *recA* for resolution of gaps introduced by UvrABC as DnaE2 would already be present. If the damage-independent production of DnaE2 (from a plasmid or in a *lexA* background) removes the requirement for *recA* this would support the gap-filling mechanism proposed by the authors. If, on the other hand, the gaps are repaired through homologous recombination, the cells would remain dependent on *recA* for gap resolution. If resolution does turn out to require *recA* even when DnaE2 is already present, it would be pertinent to test whether *recO* is also required for resolution. A requirement for both *recA* and *recO* would be strong evidence in support of a homologous recombination mechanism.

Note that if the authors chose to express DnaE2 from a plasmid, this construct could be used to complement DnaE2 function in their chromosomal null and catalytic-dead mutations, which would further solidify their further observations.

Reviewer #3:

This manuscript by Joseph et al. demonstrates that the *Caulobacter* TLS polymerase DnaE2 plays a role in NER in non-dividing cells. Using cell-biological imaging the authors show that replisome components localize on DNA in non-replicating swarmer cells after treatment with DNA damaging agents. Resolution of these foci requires SOS activation and the catalytic activity of DnaE2. These data contribute to a growing view that TLS polymerases contribute to DNA damage tolerance outside the replisome. In all the manuscript is well-written and properly supported by the data. The authors should consider the following issues:

1. I find it odd that ImuB is never discussed given that it is required for ImuC (DnaE2) mutagenesis. While I don't think it is strictly required for publication it would certainly strengthen the paper if the authors tested whether ImuB is required for the resolution of DnaN foci.

2. Figure 2C – Given the very slow kinetics in Figure 2D is the loss of a focus in 2C really representative? How do the authors ensure that the much faster kinetics seen in 2C is not due to photobleaching resulting from a much faster imaging rate compared to 2D?

[Editors’ note: further revisions were suggested prior to acceptance, as described below.]

Thank you for submitting your article "Coordination between NER and specialized polymerase DnaE2 action enables DNA damage survival in non-replicating bacteria" for consideration by *eLife*. Your article has been reviewed by 3 peer reviewers, and the evaluation has been overseen by a Reviewing Editor and Kevin Struhl as the Senior Editor. The following individuals involved in review of your submission have agreed to reveal their identity: Andrew Robinson (Reviewer #2); Robert Fuchs (Reviewer #3).

Essential Revisions:

After a substantial discussion, the reviewers agreed, that a substantial amount of new data added to the revised manuscript significantly strengthens the authors' conclusions. The Discussion section, however, needs a careful reframing.

1. The discussion between the reviewers and the editor was centered on the importance and relative frequency of the pathway in which DnaE2 participate. The authors use the term NER quite loosely, but do not actually observing all NER events, rather LPER. We agreed that it is important that the authors clarify in the Discussion that LPER is likely a minor pathway, but important when there is a high density of lesions.

2. The authors need to clearly articulate the differences between the regular NER and long patch excision repair, which likely involves DnaE2, and include references for LPER, which seem to be missing.

3. In the methods section, the authors should include the exposure times. The authors should also comment on what they think would be the shortest event that would produce a focus in their images.

*Reviewer #1:*

The additional experiments and clarifying text have substantially strengthened the manuscript. The authors have addressed my major concerns in my prior review.

I have only one comment that the authors should consider addressing:

What is the abundance of SSB-YFP relative to endogenous SSB? The authors currently remark that gaps must be >30 nt to enable SSB binding. However if there is a substantial pool of unlabeled SSB present this would suggest that the authors are likely undercounting SSB foci (especially if the gap size is on the order of single tetramer binding). Although not required, it would be interesting if the authors described the intensity of these SSB foci. For example do most foci bleach in a single step consistent with a small gap or instead are they quite bright suggesting much longer ssDNA tracks?

*Reviewer #2:*

Joseph et al., use a novel approach to investigate the roles of the DNA polymerase DnaE2 in the bacterium *Caulobacter crescentus*. By generating non-replicating cells and using live-cell imaging to observe key DNA replication/repair proteins, they make a compelling case for the existence of a novel DnaE2 activity. The authors' primary conclusion is that under non-replicating conditions, DnaE2 carries out gap-filling within intermediates of the nucleotide excision repair pathway and that this activity supports the survival of cells treated with DNA damaging agents.

Strengths

The use of non-replicating swarmer cells of *Caulobacter crescentus* enables the authors to experimentally isolate DnaE2 activities that are independent of DNA replication. The use of live-cell fluorescence imaging enables the authors to directly observe DNA replication and repair proteins as they work on the DNA. By combining the two approaches, the authors reveal a novel activity for DnaE2 that is unlikely to have been discovered in any other way. The imaging results are further supported by traditional genetics approaches.

Weaknesses

Due to technical limitations the main players – DNA gaps, UvrA and DnaE2 – are not directly observed in this study. Instead, the molecular model that the authors propose derives from observations of other DNA replication and repair proteins in cells that maintain, or lack, UvrA and DnaE2. It is important to note that the study generates significant new insight and in my opinion this shortcoming does not outweigh the strengths of this study. Additionally, a number of important mechanistic questions are discussed but have not been explored experimentally in the current study. Why would a specialized DNA polymerase required for a simple gap-filling activity? Why is RecA required for gap filling? Why is RecA so important for cell survival (much more so than DnaE2) under non-replicating conditions where no recombination is possible? There is a high likelihood that these questions will be addressed experimentally in follow-up studies.

Overall the authors are successful in identifying new phenomena that are consistent with a role for DnaE2 in gap-filling of repair intermediates generated by nucleotide excision repair. The primary contribution of the study is that it adds to a growing list of activities for DNA polymerases outside the context of the DNA replication fork – an area that has attracted considerable attention within the DNA repair community over the past five years.

There are quite a few spelling and grammar errors throughout the manuscript. The paper would benefit from careful attention to these. Additionally, in some cases the description of the methods is a little ambiguous. A clear example of this is in the description of the survival assays. As it stands it is not clear that survival was measured using non-replicating swarmer cells. It seems likely that survival was measured using swarmers, however this is not clearly stated in the Methods or Figure Captions. Some minor rewriting to improve the clarity and precision of statements throughout the manuscript would be highly beneficial.

*Reviewer #3:*

By using an interesting experimental system of *Caulobacter* swarmer cells, the authors address the mechanism of DNA lesion processing in the absence of replication. Lesions generated by MMC and UV light are known to be essentially repaired by NER; in *Caulobacter*, as well as in *E. coli* and other model bacteria, NER entails several steps, (i) lesion recognition and excision by the UvrABC complex, (ii) gap-filling and (iii) ligation. The work mostly focuses on the gap-filling step that is thought to be essentially dependent on DNA PolI (line 422). Surprisingly, the authors' work points to the involvement of polymerase DnaE2 (ImuC) in lesion processing under these non-replicative conditions. DnaE2 was previously shown to be the main polymerase involved in induced mutagenesis in *Caulobacter* and has thus been categorized as a TLS polymerase. Nevertheless, increasing numbers of reports have shown that, polymerases initially discovered as being involved in lesion bypass and mutagenesis, also participate in other metabolic processes and are now more commonly called specialized DNA polymerases.

General comment: the treatments used in the present paper induce thousands of DNA lesions per bacterial genome (i.e. at 75J/m2 of UV light the number of dimers is ≈3000/bacterial genome). Given the huge number of potential repair events, I wonder if their visualization in the form of fluorescent foci (as aimed in this paper) is realistic? The situation is different for ongoing replication forks for which there are only a few per cell.

This paper requires the following major revisions:

1. The main objective of the paper is to study NER following MMC and UV damage under non-replicative conditions. As mentioned above, DNA synthesis during NER is commonly considered to be performed by PolI. While the paper is focused on the potential involvement of DnaE2, it is difficult to understand why the authors did not investigate the localization of PolI and DnaE2 in parallel? In contrast, the authors use fluorescent fusion strains to visualize DnaN, SSB, DnaE and HolD, that appear all to be less central to NER than DnaE2 and PolI.

2. Role of SSB: the authors conclude that SSB binds to NER-generated gaps; classical NER gaps (12-13bp) are way too short to support binding of even a single SSB tetramer (30nt).

However, a pathway, referred to as Long Patch Excision Repair (LPER), has been described in *E. coli* (Cooper, 1882; Cooper and Hanawalt, 1972). LPER patches are at least 1500 nt long but represent only about 1% of the classical NER events. The mechanism of formation and the biological significance of these long patches has not been elucidated yet. The size of the patches generated during LPER are compatible with SSB filament formation. It is thus possible that these LPER events are the events that specifically require DnaE2 and the ones that are detected in the present paper. This would also explain why there are only few NER-dependent foci/cell (as noted by the authors on line 480). Additional experimentation is necessary to confirm or exclude the LPER hypothesis.

3. Throughout the paper, the authors appear to consider MMC and UV irradiation as being similar DNA damaging agents. This is probably not accurate, as MMC induces ICL's to a much larger extent than UV-light. This point is seen in Figure 2 suppl.1 A and D: compared to wt, dnaE2 is almost equally sensitive to UV light while it is highly sensitive to MMC. This may point to the specific involvement of DnaE2 in ICL repair.

Cooper, P. K. (1982). Characterization of long patch excision repair of DNA in ultraviolet-irradiated *Escherichia coli*: an inducible function under rec-lex control. Molecular and General Genetics : MGG, 185(2), 189-197.

Cooper, P. K., and Hanawalt, P. C. (1972). Heterogeneity of patch size in repair replicated DNA in *Escherichia coli*. Journal of Molecular Biology, 67(1), 1-10.

The possibility that DnaE2 is specifically restricted to filling-in the LPER gaps should be considered. In contrast, the 99% of normal NER patches are filled in by PolI.

Comparison between DnaE2 and PolI fusion strains may show a huge difference in foci formation.

---

## [Author Response]

[Editors’ note: the authors resubmitted a revised version of the paper for consideration. What follows is the authors’ response to the first round of review.]

Reviewer #1:My overall comment is that the data do support neither the title nor the abstract of the paper. In the title "bacterial DNA damage survival" is mentioned while no damage survival data are provided. In the abstract, the authors say they provide evidence for "replication-independent TLS". At the best their data show that a TLS polymerase, namely DnaE2, participate in NER gap-filling which does not mean that DnaE2 performs a lesion bypass. Indeed, a gap generated during NER does in principle not contain a lesion except in the rare situations of closely spaced lesions. If this were indeed the case the authors should observe a quadratic dose response. The involvement of DnaE2 during NER does not mean that DnaE2 is performing Translesion synthesis (TLS). There are more and more situations describing the involvement of TLS polymerases in transactions that do not involve lesion bypass per se. For example, the involvement of Pol kappa has been described many years ago during NER-gap filling in yeast and in human cells without bona fide lesion bypass (see ref below). To assess genuine lesion bypass the authors would need to monitor induced mutagenesis that is the signature of TLS.

We are grateful to Dr. Fuchs for thorough and constructive review of our work. Based on the feedback, we have made extensive revisions to our original manuscript. We address all comments individually below. Here we address three key points raised by the reviewer:

A). Role of specialized DNA polymerases in gap-filling on NER-generated substrates

We would like to start by thanking Dr. Fuchs for raising an important criticism about our usage of the term ‘TLS’ and apologise for the issues stemming from lack of clarity in our writing and interpretation of results. We originally used the term TLS to describe DnaE2 action as it is the only known mutagenic polymerase involved in lesion bypass in the limited number of organisms it has been studied in. However, we agree that the usage of this term can be misleading because it invokes the idea of potential mutagenic lesion bypass. Indeed, the focus of our study is to highlight the gap-filling role of this polymerase on NER-generated substrates.

During the course of our revisions, we came across this excellent review article published during the time of our manuscript revisions (Fuji and Fuchs, Nov 2020, MMBR). Here the authors address issues about TLS definitions, and the need to expand this definition to encompass the idea of ‘specialized polymerases’ in bacteria. Fuji and Fuchs make an important argument that error-prone polymerases can function beyond their canonical role of replication-associated lesion bypass. Thus, a more expansive definition of their action must be considered. For this, the term ‘specialized polymerases’ could be employed to consider error-prone polymerases that may have roles other than replication-associated TLS. Thus, in our revised manuscript, we now use the term ‘specialized polymerase’ or ‘low fidelity polymerase’ to refer to DnaE2 activity in gap-filling, irrespective of the possibility of mutagenic lesion bypass. The steps that result in gap-filling mediated by DnaE2 in non-replicating cells is the central focus of this manuscript.

The aspect of resolving the function of DnaE2 in ‘gap-filling alone’ or ‘lesion bypass + gap-filling’ itself is far more complex and this is not a central point in our currently reported investigations. It would be exciting to resolve the specific function of DnaE2 in gap-filling vs mutagenic lesion bypass. However, the complexity of experiments required to do so precludes our ability to conclusively use our current assay design of non-replicating *Caulobacter* swarmer cells to answer this question (expanded further in point B). We thus feel that to be able to systematically address the specific point of mutagenic potential of DnaE2 activity, an independent comprehensive study is best done. This is indeed part of forthcoming investigations conducted by a PhD student in the lab. Hence, the current manuscript focusses on the gap-filling role of DnaE2. To the best of our knowledge, in case of DnaE2 (a highly conserved but poorly studied error-prone polymerase) (Wu et al., 2012) ours is one of the first in vivo, cell biological studies to probe its activity in a cell cycle independent context, providing direct evidence for DnaE2 function on NER-generated substrates in non-replicating cells.

We have now significantly rewritten the text (and modified the title and abstract) to refer to gap-filling and avoid speculative implications of DnaE2 action on mutagenesis. Based on the feedback and the literature evidence (including the eukaryotic studies highlighted by Dr. Fuchs), we have rewritten all sections of the text to use the term ‘specialized polymerase’ or ‘low fidelity polymerase’ to describe DnaE2 function in gap-filling, avoiding reference to TLS in our context (for example, L73-74 and other sections of the manuscript). We also provide data on dose-dependent effect of DnaE2 action on replisome persistence as well as survival in non-replicating cells (Figure 4—figure supplement 1C, Figure 5D) (expanded further in points B and C).

B). Why is there need for specialized polymerase during replication-independent NER?

Indeed, a current challenge of our study is to understand why DnaE2 action is required for such gap filling. Given persistence of the replicative polymerase (DnaE) in the absence of DnaE2 at ssDNA gaps (suggesting inability of DnaE to complete synthesis across the same) (Figure 4C), and requirement of synthesis activity of DnaE2 (Figure 4D), in our original manuscript, we had speculated on a role of specific NER-generated substrates, such as COLs, in requiring DnaE2-mediated synthesis.

Based on feedback from the reviewer, we performed additional experiments at various doses of MMC damage and observe a dose-dependent effect of DnaE2 action on replisome persistence (Figure 4—figure supplement 1C) as well as cellular survival (measured by estimating viable count of the population) (point C and Figure 5D). We also conducted assays to measure rates of rifampicin resistance generation (as a measure for mutagenesis) in our experimental setup (Figure 4—figure supplement 1G). We find that increase in mutagenesis in response to damage (independent of presence or absence of replication) arises from DnaE2 activity. However, rifampicin resistance assays are technically very challenging in the non-replicating swarmer cell setup (which requires us to conduct several DnaA depletions and density gradient synchronizations across large culture volumes), and it is also difficult to reliably disentangle the confounding effects of replication during the outgrowth period. Our current setup is unique in that it allows us to be 100% sure that there is no ongoing replication in our system and that cells cannot undergo recombination due to absence of a second chromosome (Figure 1A). Thus, given the complexity of the experimental design and challenges associated with conducting the assays to measure mutagenesis with reliability specifically in non-replicating conditions, we prefer to continue to maintain focus on gap-filling by DnaE2 and the survival advantage it confers to non-replicating cells (toning down our reference to COLs). To resolve above raised question in a comprehensive and reliable manner, extensive work on DnaE2 action (which is poorly studied, unlike *E. coli* error-prone polymerases) would need to be conducted. This is outside the scope of our current study and does not affect the central observations reported here.

We include new data on replisome foci dynamics as well as cellular survival (measured via CFU assays) in the presence and absence of DnaE2 across a range of MMC doses (Figure 4—figure supplement 1C, Figure 5D and L363 to L383). We also include results from rifampicin resistance assays for calculating mutagenesis, and provide explanation about the difficulty to disentangle the impact of the replication-proficient outgrowth period on mutagenesis observed under damage (Figure 4—figure supplement 1G and L300 to L307). We have ensured to tone-down our speculation on the impact of COLs on the pathway we describe in our current study (and removed the speculative model figure from the manuscript) (L427 to L434). We have maintained the focus on the gap-filling role of DnaE2 and its impact on survival.

C). Impact of gap-filling by specialized polymerases on bacterial DNA damage survival:

In our original manuscript, we used live cell imaging-based read-outs to draw direct conclusions on the importance of coordination of NER with replication-independent DnaE2 activity in enabling survival under DNA damage. In Figure 5C and Results section starting L329 we provided evidence for a survival advantage in cells where gap-filling has been completed. For this, we specifically scored survival as the ability of a cell to undergo one or more divisions after release into replication-permissive conditions, with cell division proficiency being a reliable indicator of cells where repair has been completed (Vickridge et al. 2017). We found that cells that engaged in repair had a 4-fold survival advantage over cells that either lacked DnaE2 or were not given the time to repair in non-replicating phase (Figure 5C). We supported this experiment with evidence on cell length elongation, a hallmark of persistent DNA damage and continued SOS response activation (Wehrens et al., 2018). We showed that cell length continued to increase (without intermittent divisions) in cells that either lacked DnaE2 or were not given a window of recovery to complete gap-filling before release into replication-permissive conditions. However, in cells that were able to complete gap-filling, cell length was restored close to a wild type distribution and cell cycle progression as well as division completion was initiated soon after release into replication-permissive conditions (Figure 5—figure supplement 1A and L348 to L362).

Based on feedback from Dr. Fuchs, we have now additionally carried out survival assays via CFU measurement to reinforce the conclusions outlined above. We now conduct survival analysis (via measuring viable cell count) across a range of MMC doses (0.125, 0.25, 0.5 and 0.75 µg/ml of MMC; further explanation of the doses chosen are given in the next point). In the absence of recovery or DnaE2 action, we find that viable cell count is significantly compromised (Figure 5D and L363 to L383) and no survival advantage is observed when non-replicating cells are released into replication-permissive conditions. In line with our cell biological read-outs, this assay supports the conclusion that enhanced survival at higher doses of DNA damage in non-replicating conditions is dependent on DnaE2 action during recovery. We speculate that these effects are likely observed because of incomplete gap-filling as seen by persistence of SSB foci in conditions where DnaE2-mediated synthesis has been compromised (see discussion L504 onwards).

We include results from survival assays conducted via measurement of viable cell count across various doses of MMC treatment (L363 to L383 and Figure 5D). In addition, we explain our cell biological read-outs for survival with clarity (L329 to L362, Figure 5C and Figure 5—figure supplement 1A).

Substantive concerns:1. Proper description of the level of damage induced by MMC and UV irradiation should be provided. At the best I can tell the authors only used one dose of MMC and one dose of UV. There needs to be experiments showing the physiological effect of MMC treatment or of UV irradiation such as survival (as claimed in the title) or mutagenesis.

We thank Dr. Fuchs for raising this point about damage doses used. We have now conducted key experiments of across several doses of DNA damage.

In our original manuscript, we had used a sub-inhibitory dose of two distinct DNA damaging agents based on literature evidences (MMC and UV; as described previously in Galhardo et al., 2005 and Modell et al., 2011), which elicited similar responses in our assays. In addition, we had tested two doses of MMC and UV in Figure 2—figure supplement 1F, Figures 4—figure supplements 1C and 1D. We had done so based on cell viability assays to ensure that the doses used show essentiality for DnaE2, but non-lethal for wild type. Given the various types of damages (on the guanine residue) caused by MMC, we are uncomfortable to provide estimate of lesion numbers as this may not be accurate. Hence, we rely on estimating the viable cell count for wild type and dnaE2 deleted cells and choose doses of damage where DnaE2 essentiality is observed, but wild type growth is not severely affected (with similar patterns of cell growth for both UV and MMC doses of damage, Figure 2—figure supplements 1A and 1D). Furthermore, unlike *E. coli* UmuDC that is thought to preferentially act on UV damage, DnaE2 (which is present in all organisms that lack UmuDC, with mostly GC-rich genomes) preferentially acts on MMC-induced damage. Thus, we specifically focus our study on MMC as it creates the damage substrates that DnaE2 specifically acts on (most likely damage on guanine, as proposed by others and supported by our preliminary unpublished data as well), utilizing UV damage only as a corroborating/ secondary source of damage.

In our revised manuscript, we now provide viable cell count data (for steady-state cells with and without DnaE2) across a range of MMC/ UV treatment used to identify the doses of damage treatment used in our study (Figure 2—figure supplement 1A and 1D), where DnaE2 becomes increasingly essential for survival. Using four doses of MMC (0.125, 0.25, 0.5 and 0.75 µg/ml) and two doses of UV damage (75 and 150 J/m2), we carry out cell biological assays to measure replisome focus formation as well as dissociation during recovery in non-replicating conditions. We also assess persistence of foci at all doses of damage in case of dnaE2 deletion (Figure 4—figure supplements 1C and 1D). Finally, we use genetic read-outs to assess survival via viable cell count assay at all doses of MMC damage in the presence and absence of DnaE2. We find that at low damage dose (0.125 µg/ml MMC), ~10% cells show replisome localization in response to damage and this number reduces to ~2% in a DnaE2-independent manner. Similarly, in the viable cell count assay as well, at low dose of damage survival appears to be independent of DnaE2 action. However, at higher doses of damage, both replisome dissociation and survival advantage in non-replicating conditions is dependent on DnaE2 (Figure 5D). Thus, we now report a dose-dependent effect of DnaE2 on replisome persistence as well as cellular survival in non-replicating conditions.

We include characterization of DNA damage doses used in this study in Figure 2—figure supplements 1A and 1D. We also provide a description of damage doses used and literature evidence for preference of DnaE2 action on MMC damage (L166 to L170 and L197 to L199). We further include data assessing effect of DnaE2 action on replisome foci persistence and impact on survival at multiple doses of damage (Figure 4—figure supplements 1C and 1D, Figure 5D).

2. All data are based on fluorescence imaging. Other methodologies (genetics…..) should be implemented to reinforce the study.

We now include other read-outs to support of the key conclusion of our work (see points discussed above, as well as experiments in response to Reviewer 2 and 3).

Ogi T, Lehmann AR. The Y-family DNA polymerase kappa (pol kappa) functions in mammalian nucleotide-excision repair. Nat Cell Biol. 2006; 8: 640-642. https://doi.org/10.1038/ncb1417 PMID: 16738703Lehmann AR. New functions for Y family polymerases. Mol Cell. 2006; 24: 493-495. https://doi.org/10.1016/j.molcel.2006.10.021 PMID: 17188030

We have now included relevant references in our revised manuscript.

We are grateful to Dr. Fuchs for the insightful review of our work. We would be happy to provide additional clarifications and experiments as advised.

Reviewer #2:Joseph et al. report a single-molecule fluorescence microscopy study that reveals interesting new links between nucleotide excision repair and the translesion synthesis DNA polymerase DnaE2 in non-replicating *Caulobacter crescentus* cells. By analysing non-replicating cells, the authors are able to examine replication/repair activities that occur outside of the context of the replication fork. The data provide strong support for a functional link between NER and TLS occurring under these conditions. The mechanism they propose is reasonable, however I think that an alternative mechanism that involves homologous recombination would fit their data equally well. Decoupling DnaE2 expression from the RecA*-mediated SOS response should allow the authors to distinguish between the two mechanisms.

We are grateful to Dr. Robinson for the positive assessment of our manuscript and for the helpful feedback and suggestions. Dr. Robinson has raised an important point with regards to the mechanism of DnaE2-mediated gap-filling (recombination vs direct synthesis on NERgenerated substrates). We have addressed this comment below.

1. It seems possible to me that the gaps created by the Uvr proteins might be repaired via homologous recombination, as opposed to gap-filling. The recruitment of SSB, HolB, DnaN, and DnaE to the repair intermediates, and the strong dependence on *recA* would be consistent with this idea. The requirement for DnaE2 could be consistent with a role in recombination, such as D-loop extension. As the authors point out, the footprint of SSB is larger than the typical gaps produced by UvrABC, so SSB foci should not form on these short gaps. SSB foci might instead form if the initial gaps are enlarged in preparation for homologous recombination (perhaps through the actions of RecQ and RecJ, as has been reported in *E. coli*). The authors also report robust induction of the SOS response. This requires the formation of RecA* nucleoprotein filaments, such as those formed in preparation for homologous recombination. It might be possible for RecA* to form on short gaps produced by UvrABC, but it seems more likely (and consistent with the formation of SSB foci) that RecA* is forming on gaps that have been enlarged for homologous recombination. The observation that *recA* is not required for the formation of DnaN foci is consistent with the notion that clamps are loaded at gaps created by UvrABC. It remains formally possible, however, that clamps are loaded at recombination intermediates in the presence of RecA, and at SSB-coated gaps in its absence. Both scenarios would lead to focus formation.2. Decoupling the production of DnaE2 from the formation of RecA* should allow the authors to distinguish between their proposed gap-filling mechanism and a homologous recombination mechanism. They could do this by expressing DnaE2 from a plasmid, or by introducing a *lexA* null mutation to make the cells SOS constitutive. Each approach has advantages and disadvantages, so ideally both would be tested. If a gap-filling mechanism is at play, the cells should no longer require *recA* for resolution of gaps introduced by UvrABC as DnaE2 would already be present. If the damage-independent production of DnaE2 (from a plasmid or in a *lexA* background) removes the requirement for *recA* this would support the gap-filling mechanism proposed by the authors. If, on the other hand, the gaps are repaired through homologous recombination, the cells would remain dependent on *recA* for gap resolution. If resolution does turn out to require *recA* even when DnaE2 is already present, it would be pertinent to test whether *recO* is also required for resolution. A requirement for both *recA* and *recO* would be strong evidence in support of a homologous recombination mechanism.Note that if the authors chose to express DnaE2 from a plasmid, this construct could be used to complement DnaE2 function in their chromosomal null and catalytic-dead mutations, which would further solidify their further observations.

We thank Dr. Robinson for highlighting this possibility. We realize that we should have discussed the same in detail as this was an important consideration we had already taken into account while designing our assay system. In our assays, we ensured that the second copy of the chromosome is unavailable in cells during DNA damage recovery (all cells have only 1n chromosome content; Figure 1A). As stated in results L130, L134, and Figure 1A in our revised manuscript, we isolate *Caulobacter* swarmer cells that have single chromosomes. Thus, the possibility of recombination is ruled out in this system, suggesting strongly that the TLS polymerase, DnaE2 alone is sufficient to carry out gap-filling on NER-generated substrates. We have now clearly considered this possibility in the main text and provided evidence ruling the same out (L251 onwards).

To further support the idea that DnaE2 directly participates in gap-filling on NER-generated substrates, we also conducted our experiments in cells deleted for *recN*. This repair protein is essential for RecA-mediated recombination repair in *Caulobacter* (Badrinarayanan et al., 2015). We find that neither association nor dissociation of the β-clamp is perturbed in the absence of *recN* (with dynamics as seen in wild type conditions), ruling out a role for recombination in the process described in our manuscript. These data are now included in the manuscript (L257 and Figure 4—figure supplement 1A)

In addition to the above evidences, we also considered the experimental suggestions made by Dr. Robinson. The *lexA* constitutively active mutant prevents the ability for us to conduct synchronization experiments to isolate non-replicating swarmer cells with a single chromosome. Thus, we prefer not to conduct this experiment as it would make our experimental system noisy. Our current setup is unique in that it allows us to be 100% sure that there is no ongoing replication in our system and that cells cannot undergo recombination due to absence of a second chromosome. Secondly, while the complementation experiment would be a nice method to additionally rule out a role for recombination, complementation of DnaE2 alone would be insufficient. Under the SOS response, three components for *Caulobacter* TLS are expressed (ImuA, ImuB and ImuC (DnaE2)), all of which are essential for successful synthesis by DnaE2. Indeed, in our revised manuscript we provide evidence for the same (based on feedback from Reviewer 3) (Figure 4D). Thus, we would need to inducibly express all three components at appropriate ratios. This has been extremely challenging till date (from ours and others labs, likely given the large size of the entire operon and presence of one regulatory RNA whose function is currently unknown).

Indeed, for the current revisions as well we attempted to construct such a tool, but faced severe technical challenges in the process. We feel unfortunate, but are confident that our swarmer cell setup (with a single chromosome) as well as *recN* deletion experiments address the point raised by Dr. Robinson.

We now explain our non-replicating, single chromosome, swarmer cell setup with clarity (Figure 1A and L130 onwards) to rule out role for recombination in the DnaE2-mediated gap-filling process (L251 onwards). We also provide data for cells showing recovery in the absence of the recombination protein RecN (L254 to 258 and Figure 4—figure supplement 1A).

Reviewer #3:This manuscript by Joseph et al. demonstrates that the *Caulobacter* TLS polymerase DnaE2 plays a role in NER in non-dividing cells. Using cell-biological imaging the authors show that replisome components localize on DNA in non-replicating swarmer cells after treatment with DNA damaging agents. Resolution of these foci requires SOS activation and the catalytic activity of DnaE2. These data contribute to a growing view that TLS polymerases contribute to DNA damage tolerance outside the replisome. In all the manuscript is well-written and properly supported by the data. The authors should consider the following issues:1. I find it odd that ImuB is never discussed given that it is required for ImuC (DnaE2) mutagenesis. While I don't think it is strictly required for publication it would certainly strengthen the paper if the authors tested whether ImuB is required for the resolution of DnaN foci.

We thank the reviewer for raising this point. We did not discuss the role of ImuB as we had considered that characterization of the same was outside the scope of our current study. However, based on feedback from the reviewer, we have now conducted our experiments in cells lacking *imuB* as well. We find that, as in the case of DnaE2, ImuB is also essential for β-clamp dissociation during DNA damage recovery (Figure 4D and L308 to L316). In addition, we highlight the importance of ImuB for DnaE2 function (L66, L309 and L476). We have also confirmed physical interaction between DnaN and ImuB via a bacterial two-hybrid assay and would be happy to provide these data if additionally required.

2. Figure 2C – Given the very slow kinetics in Figure 2D is the loss of a focus in 2C really representative? How do the authors ensure that the much faster kinetics seen in 2C is not due to photobleaching resulting from a much faster imaging rate compared to 2D?

This is an important point raised by the reviewer. Unfortunately, the experimental regimes are not directly comparable between Figure 2C and Figure 2D. Thus, given that these data are not central to our current manuscript, we have removed this figure panel (2C) from the revised manuscript presently as it does not contribute significantly to the conclusions of the study. We maintain consistent focus on the analysis over longer timescales (Figure 2 and elsewhere throughout the manuscript).

[Editors’ note: what follows is the authors’ response to the second round of review.]

Essential Revisions:After a substantial discussion, the reviewers agreed, that a substantial amount of new data added to the revised manuscript significantly strengthens the authors' conclusions. The Discussion section, however, needs a careful reframing.

We are very grateful for the thoughtful consideration of our work and for guidance with regards to essential revisions we need to make to our manuscript.

1. The discussion between the reviewers and the editor was centered on the importance and relative frequency of the pathway in which DnaE2 participate. The authors use the term NER quite loosely, but do not actually observing all NER events, rather LPER. We agreed that it is important that the authors clarify in the Discussion that LPER is likely a minor pathway, but important when there is a high density of lesions.

We thank the Editor and Reviewers for highlighting this critical point. We agree that we are not observing all NER events and that it is important to consider the possibility that DnaE2 is required only for a subset of NER events that involves gap-filling across long patches (likely at higher doses of damage). Hence, based on this feedback, we have revised our Discussion section to discuss this point with clarity (L433 to L448). We include LPER as a possible scenario where DnaE2 activity may be required, in addition to the alternate model that would involve generation of long ssDNA gaps at problematic intermediates of NER. We state that a limitation of our study is that we are not tracking all NER events (a significant proportion of which could be mediated via gap-filling by PolI across short patches of ssDNA) and that the mechanism by which long ssDNA gaps are generated from a subset of NER events remains unclear (L449 to L453). Our future efforts are aimed at understanding why there is a need for specialized polymerase activity on some NER-generated substrates at higher doses of damage.

2. The authors need to clearly articulate the differences between the regular NER and long patch excision repair, which likely involves DnaE2, and include references for LPER, which seem to be missing.

We have now included reference to LPER. In the introduction (L42) and results (L227) sections we briefly mention the possibility of LPER. In the discussion (L420 onwards), we consider the differences between regular NER and long patch repair and discuss the specific involvement of DnaE2 in long patch repair at higher doses of damage.

3. In the methods section, the authors should include the exposure times. The authors should also comment on what they think would be the shortest event that would produce a focus in their images.

We have revised this section of the manuscript to provide information on exposure times (L571).

We apologise if we have misunderstood the query with regards to focus formation in our experimental regime. We address this comment in two parts and would be happy to provide additional clarifications if required. (1) With regards to stability, we are confident that the foci we observe are likely bound molecules in stable structures, as reported for replisome components in previous studies pertaining to DNA replication and repair in *E. coli* (Soubry et al., 2019) based on the following observations: In our experiments, we find that varying exposure times (50-400 ms) do not influence the number of foci observed per cell under damage (and no foci are detected in the absence of damage). In support, we also note that once formed, a localization is observed for 6 min on average before dissociation. Since our imaging setup does not have the resolution to allow us to comment on binding kinetics with precision (as we are not detecting single molecule binding events and transient binding events), we prefer not to make conclusions about the same. We would also like to highlight that focus dissociation was repair-dependent (as assessed by lack of dissociation of individual foci in absence of *dnaE2*). (2). With respect to duration of damage events, we did conduct our experiments at short exposures to UV damage as well. We observed DnaN foci at 30 J/m^2^ UV exposure (dose used in Henrikus et al., 2018; *E. coli* study tracking TLS activity under damage via live cell imaging) and 50 J/m^2^ (dose used in Soubry et al., 2019; *E. coli* study tracking replisome activity under damage via live cell imaging). As anticipated, the percentage of cells with foci increased with increasing dose of damage. For the experiments reported in our current work, we chose a regime of damage where activity of DnaE2 is required and wild type cells can repair damage (with minimal impact on cell death).

References

Henrikus, S. S., Wood, E. A., McDonald, J. P., Cox, M. M., Woodgate, R., Goodman, M. F., van Oijen, A. M., and Robinson, A. (2018). DNA polymerase IV primarily operates outside of DNA replication forks in *Escherichia coli*. *PLoS Genetics*, *14*(1), e1007161. https://doi.org/10.1371/journal.pgen.1007161

Soubry, N., Wang, A., and Reyes-Lamothe, R. (2019). Replisome activity slowdown after exposure to ultraviolet light in *Escherichia coli*. *Proceedings of the National Academy of Sciences*, 201819297. https://doi.org/10.1073/pnas.1819297116